# Air quality simulations for London using a coupled regional-to-local modelling system.

Christina Hood[1], Ian MacKenzie[2], Jenny Stocker[1], Kate Johnson[1], David Carruthers[1], Massimo Vieno[3], Ruth Doherty[2]

[1] Cambridge Environmental Research Consultants, Cambridge, CB2 1SJ, United Kingdom
[2] School of GeoSciences, The University of Edinburgh, Edinburgh, EH8 9XP, United Kingdom
[3] Centre for Ecology and Hydrology, Edinburgh, EH26 0QB, United Kingdom

*Correspondence to*: Christina Hood (chood@cerc.co.uk)

**Abstract.** A coupled regional-to local modelling system comprising a regional chemistry-climate model with 5 km horizontal resolution (EMEP4UK) and an urban dispersion and chemistry model with explicit road source emissions (ADMS-Urban) has been used to simulate air quality in 2012 across London. The study makes use of emission factors for $NO_x$ and $NO_2$ and non-exhaust emission rates of $PM_{10}$ and $PM_{2.5}$ which have been adjusted compared to standard factors to reflect real-world emissions, with increases in total emissions of around 30% for these species. The performance of the coupled model and each of the two component models is assessed against measurements from background and near-road sites in London using a range of metrics concerning annual averages, high hourly average concentrations and diurnal cycles. The regional model shows good performance compared to measurements for background sites for these metrics, but under-predicts concentrations of all pollutants except $O_3$ at near-road sites due to the low resolution of input emissions and calculations. The coupled model shows good performance at both background and near-road sites, broadly comparable with that of the urban model which uses measured concentrations as regional background, except for $PM_{2.5}$ where the under-prediction of the regional model causes the coupled model to also under-predict concentrations. Using the coupled model, it is estimated that 13% of the area of London exceeded the EU limit value of 40 µg m$^{-3}$ for annual average $NO_2$ in 2012, whilst areas of exceedences of the annual average limit values of 40 µg m$^{-3}$ and 25 µg m$^{-3}$ for $PM_{10}$ and $PM_{2.5}$ respectively were negligible.

KEY WORDS air quality, modelling, atmospheric chemistry, dispersion, London, EMEP4UK, ADMS-Urban

## 1 Introduction

Poor air quality has long been recognised as having adverse effects on health. Particulate pollution in the UK has been assessed as causing a loss of life expectancy from birth of approximately six months (COMEAP, 2010), while air pollution in the WHO European Region was estimated to cause 600 000 premature deaths in 2010 (WHO, 2015). Improved

understanding of these health effects requires additional information about air quality, especially in urban areas where high pollutant concentrations coincide with high population densities.

Continuous air quality measurements, for example from the UK Automatic Urban and Rural Network (AURN, Defra 2017), are typically carried out at a limited number of fixed locations in an urban area and are expected to be representative of 'several square kilometres' for urban background locations (EC Directive, 2008). In addition, short-term intensive campaigns making use of specialist monitoring equipment, as for example carried out for the ClearfLo project (Bohnenstengel et al., 2015), are of great value for detailed assessment of model performance and underlying processes, whilst sampling equipment can also be carried by moving vehicles or individuals for short-term detailed studies. In contrast to measurements, air quality or atmospheric chemistry transport models, evaluated with the above data, allow pollutant concentrations to be simulated with complete spatial-temporal coverage leading to detailed calculations of population exposure (Smith et al. 2016).

Air quality models require accurate input emissions data to make reliable predictions of ambient concentrations. However in the last decade, it has become clear that measured $NO_x$ and $NO_2$ concentrations have not decreased as fast as would have been anticipated from published emission factors (Carslaw et al., 2011). Several measurement techniques for direct assessment of on-road tailpipe emissions, as reported by Carslaw and Rhys-Taylor (2013) and O'Driscoll et al. (2016), have confirmed differences from the official emissions estimates (EFT, Defra 2016). In-service emissions performance evaluation of Euro 6/VI vehicles (Moody and Tate, 2017) indicated that whilst Euro VI heavy duty vehicles and Euro 6 petrol light duty vehicles are performing broadly as predicted, Euro 6 diesel light duty vehicles emit $NO_x$ at rates exceeding the published data, by factors of up to 4.5.

There is further uncertainty in the rates of particulate emissions from road vehicles due to wear of tyre, brake and road surfaces and resuspension of pre-existing particulates (Thorpe and Harrison, 2008). Particulate exhaust emissions have decreased considerably in recent years, primarily due to the introduction of diesel particulate filters, so the relative contribution of non-exhaust $PM_{10}$ and $PM_{2.5}$ to total traffic emissions is now considerable, of the order of half of the total 'exhaust' (Grigoratos and Martini, 2014).

Atmospheric chemistry models that simulate air quality vary in complexity in terms of the scales and processes represented. Global and regional models use gridded emissions data to calculate transport and chemistry over global or regional modelling domains, such as EMEP/MSC-W (EMEP, Simpson et al., 2012) used in this study, CAMx (Ramboll Environ, 2016), CMAQ (Byun and Schere, 2006) and WRF-Chem (Grell et al., 2005). Models on a smaller scale apply detailed transport and fast chemistry processes to individual sources, such as ADMS-Urban (Owen et al., 2000) also used in this study, the US EPA model AERMOD (Cimorelli et al., 2004) and CAR-FMI (Kukkonen et al., 2001).

Global and regional air quality models typically use detailed chemistry schemes whilst urban models typically only represent fast chemistry, such as $O_3$-$NO_x$ chemistry which is relevant for pollution concentration gradients across urban areas. Some

hybrid process-statistic based approaches have also been developed, where measured concentration data are used to constrain modelled concentrations in order to reduce uncertainties, for example those described in Stedman et al., 2001 and Sokhi et al., 2008.

Urban dispersion models typically use measured upwind rural concentrations to represent long-range transport. This is a successful approach for modelling historic periods but has limited applicability for assessing future scenarios, including those related to climate change or the local effects of regional emissions changes. The use of a limited number of upwind monitoring sites can also make the 'upwind' concentration data less representative, and does not allow for variations over a large urban area. An alternative method (Stocker et al. 2012, 2014) is to combine regional modelling with urban local modelling in order to take into account both short-range and long-range transport and chemistry effects, whilst avoiding "double-counting" the gridded and explicit emissions. The balance between regional and local influences differs according to the pollutant lifetime. For example, concentrations of ozone and particulates, which have lifetimes at the surface of days to weeks, are strongly influenced by regional emissions and transport, whereas concentrations of $NO_2$, with a surface lifetime of around 1 day, are primarily related to the dispersion and chemical transformation of local emissions.

The overall methodology for the detailed evaluation of air pollution concentrations across London for 2012 using a coupled regional-urban model is described in Sect. 2 with details of the measurement data, models, emissions and statistical parameters. Sect. 3 gives the results of the model evaluation against measured concentrations while Sect. 4 discusses the results in relation to air quality in London and the different modelling methods.

## 2 Methods

This paper presents a detailed evaluation of air pollution concentrations across London for 2012 using a coupled regional-urban model (described in Sect. 2.4), which comprises a regional version of the EMEP atmospheric chemistry transport model, EMEP4UK (Sect. 2.2), and the ADMS-Urban local dispersion and chemistry model (Sect. 2.3). The coupled model is evaluated alongside the stand-alone implementations of the two underlying models. The evaluation exercise compares hourly modelled concentrations of $NO_x$, $NO_2$, $O_3$, CO, $PM_{10}$, and $PM_{2.5}$ with measured hourly concentrations from up to 42 automatic monitoring sites within Greater London for the year 2012, described in Sect. 2.1. The model simulations use road vehicle emissions of $NO_x$, $NO_2$ and particulates which have been adjusted in line with real-world emissions measurements. The emissions data, including the raw 2012 emissions, adjustments and time variation, are described in Sect. 2.5. Definitions of the statistics used for model evaluation are given in Sect. 2.6.

## 2.1 Measurement Data

The monitoring sites selected for the model evaluation were those from the London Air Quality Network (LAQN, Mittal et al., 2017) located within Greater London that had at least 70% data capture of hourly data during 2012 for $PM_{10}$, $PM_{2.5}$, CO or at least two of $NO_x$, $NO_2$ and $O_3$. A summary of site numbers by type and their average heights is given in Table 1; with

the exception of two background monitors at 5.5 and 10 m all monitors are between 2 and 4 m above ground. The site locations of the $NO_2$ and $O_3$ monitors are presented in Fig. 1. Note that all the map plots in this paper adopt the polar stereographic coordinate system as used in EMEP4UK, with an approximately 30° anti-clockwise rotation of axes compared to standard UK OSGB coordinates.

## 2.2 Regional-scale Modelling: EMEP4UK

EMEP4UK is a nested regional application of the EMEP MSC-W (European Monitoring and Evaluation Programme Meteorological Sythesizing Centre-West) model, focused specifically on air quality in the UK. The main EMEP MSC-W model has been widely used for both scientific studies and for policy making in Europe, with references to evaluation and application studies available in Simpson et al. (2012), Schulz et al. (2013), and at http://www.emep.int. EMEP4UK is described in Vieno et al. (2010, 2014, 2016ab); the version used here is based on EMEP MSC-W rv4.5. It uses one-way nesting from a 50 km x 50 km resolution greater European domain (standard EMEP domain) to an inner 5 km x 5 km domain which covers the British Isles and nearby parts of continental Europe, both in a polar stereographic projected coordinate system, as shown in Fig. 2. An intermediate 10 x 10 km resolution domain is used for WRF to ensure numerical stability, but is not required for the chemistry transport calculations (Vieno et al. 2010). Full details of the WRF model domains are given in Table A1. The model has 21 vertical levels extending from the surface to 100 hPa, with the lowest vertical layer 50 m thick, meaning that modelled surface concentrations represent a height of around 25 m. This is a finer vertical resolution than was available in the standard EMEP model v4.5. Hourly average output concentrations are available from each cell for the full 2012 modelling period.

The gaseous chemical scheme used in EMEP4UK in this study is the CRI-v2-R5 mechanism (Watson et al. 2008), which has 220 species and 609 reactions. Five classes of fine and coarse particles, with differing size and deposition properties, are used in EMEP4UK (Simpson et al. 2012) along with the MARS equilibrium module for gas–aerosol partitioning of secondary inorganic aerosol (Binkowski and Shankar, 1995; Simpson et al., 2012) and a treatment of secondary organic aerosol formation using the volatility basis set approach (Bergström et al. 2012, Ots et al. 2016). Dry (including stomatal) and wet deposition of gases and particles are simulated. Sixteen land cover classes are used for dry deposition modelling and for the calculation of biogenic emissions. Ozone boundary conditions for the outer European domain are based on the approach in Simpson et al. (2012), scaling a monthly climatology with clean air measurements at Mace Head. Initial and boundary conditions of all other species for the European domain are specified as fixed functions of latitude and time of year.

The chemistry transport model was driven by meteorological output from the Weather Research and Forecasting (WRF) model version 3.6.1 (Skamarock et al., 2008; NCAR, 2008) including data assimilation of 6-hourly meteorology from the European Centre for Medium Range Weather Forecasting ERA-Interim reanalysis (Dee et al., 2011). The option of input meteorological data from WRF has been developed for EMEP4UK and is used in both the European and UK domains. The

WRF configuration was as follows: Lin Purdue for microphysics; Grell-3 for cumulus parameterization; Goddart Shortwave for radiation physics; and Yonsey University (YSU) for planetary boundary layer (PBL) height. Land use categories were based on the MODIS IGBP classification. This WRF configuration is similar to that discussed in Vieno et al. (2010), where it is shown to perform well in comparison with measurements. An evaluation of the WRF-EMEP4UK modelling system against measured gaseous and particulate pollutant concentrations across the UK for 2001 to 2010 is given by Lin et al. (2017), while Ots et al. (2016) compares WRF-EMEP4UK air quality simulations with detailed measurements of secondary organic aerosols made in London during the 2012 ClearfLo campaign.

## 2.3 Urban-scale Modelling: ADMS-Urban

The Atmospheric Dispersion Modelling System (ADMS, Carruthers et al., 1994) is a quasi-Gaussian plume air dispersion model able to simulate a wide range of passive and buoyant releases to the atmosphere. The dispersion calculations are driven by hourly meteorological profiles of wind speed and direction, among other parameters, which are characterised using Monin-Obukhov length similarity theory; meteorological input data may be derived from measurements or output from a mesoscale model such as WRF. ADMS is a local dispersion model, able to resolve concentration gradients that occur in the vicinity of a range of emission source types, including point, jet, line, area and volume sources. The modelling of dispersion and chemistry for source emissions is independent of the output grid resolution.

The ADMS-Urban model has been used to simulate air quality within cities worldwide; applications include testing of emission-reduction scenarios and forecasting (Stidworthy et al., 2017). Emissions from all sources within the model domain are included, either explicitly with detailed time-varying profiles, for instance major road and industrial sources, or as grid-averaged emissions, representing diffuse sources such as those from heating and minor roads as a grid of regular volume sources, with simpler time variation.

The flow field that drives dispersion of pollutants within an urban area is inhomogeneous. On the neighbourhood scale, buildings displace the upwind wind speed profile and reduce in-canopy wind speeds. ADMS-Urban has an 'urban canopy' flow field module, which calculates wind speed and turbulence flow profiles that relate to the spatial variation of the surface roughness length, $z_0$. Locally, if street canyons are formed by densely packed tall buildings, it is important to model the complex combination of recirculating and chanelled flows. The ADMS-Urban advanced street canyon module is able to model the channelling of flow along and circulation of flow across a street canyon, to represent asymmetric street canyons, to represent the effect of pavements within a canyon and to calculate the effect of a street canyon on the surrounding area. The module has been validated extensively by comparison with measurements from monitoring networks in Hong Kong and London (Hood et al., 2014). In the present modelling the advanced street canyon option has been used for all roads in the modelling domain with adjacent buildings.

For this project 3D buildings data and road centreline locations from Ordnance Survey MasterMap (Ordnance Survey, 2014) were processed for use in ADMS-Urban, as described in Jackson et al. (2016), although using the EMEP4UK polar

stereographic projected coordinate system. The inputs to ADMS-Urban take two forms: gridded building height and density parameters for urban canopy flow field calculations; and street canyon properties for each side of explicitly modelled road sources. Fig. 3 shows the variation of average building height over the Greater London area, which is used to determine the local roughness length for flow calculations.

In this study ADMS-Urban version 4.0.4 was used for the stand-alone runs, with emissions covering the Greater London area, defined by the LAEI emissions extent. The stand-alone runs use hourly measured meteorological data from Heathrow airport (location shown in Fig. 1) for the whole domain, with valid data available for over 95% of hours in 2012. Long-range transport of $NO_x$, $NO_2$, $O_3$, $PM_{10}$, $PM_{2.5}$ and $SO_2$ is represented by hourly measured background concentrations from rural sites upwind of London in the Automatic Urban and Rural Monitoring Network (AURN, Ricardo-AEA 2013): Wicken

Fen (North of London, gaseous pollutants only); Lullington Heath (South, gaseous pollutants only); Harwell (West, all pollutants); and Rochester (East, all pollutants), providing boundary conditions to the local modelling. No monitoring data are available for CO, so a constant background concentration for CO of 220 µg m$^{-3}$ was obtained from the annual mean background map published by Defra (2013b). Output concentrations are given at hourly resolution and post-processed to calculate long-term averages as required. The modelling also uses the ADMS-Urban $NO_x$ photolytic chemistry module,

which accounts for fast, near-road oxidation of NO by $O_3$ to form $NO_2$ (Smith et al., 2017), and simple sulphate chemistry for conversion of $SO_2$ to $PM_{2.5}$ and $PM_{10}$. The options for flow over complex terrain (Carruthers et al., 2011), and gaseous and particulate wet and dry deposition are not used in this study.

## 2.4 Coupled Regional-to-Urban scale model

At short times after release of a pollutant from a source, concentration gradients due to releases from that source are high and

a street-scale resolution model such as ADMS-Urban is needed to capture the fine details of dispersion and fast chemistry, for instance at roadside locations. At longer times after release, pollutant concentration gradients are reduced by mixing and a gridded regional model that accounts for long-range transport and detailed chemical transformations simulates these processes adequately. These models may be coupled within a single system. However, the computational linkage process is non-trivial in order to avoid double-counting of emissions and to ensure that the chemical processes are accounted for at all

time scales.

The underlying concept for coupling the regional and urban scale models, described in Stocker et al. (2012), is to use the local urban model to represent the initial dispersion of emissions up to a mixing time, typically one hour, after which the emissions are considered well-mixed on the scale of the regional model grid. In general, Gaussian plume models such as ADMS-Urban treat plumes as continuing for all time, but within the coupled system the calculations are truncated at the

mixing time $\Delta t$. An ADMS-Urban run with gridded emissions, limited to the mixing time ($C_{grid}(\Delta t)$), is used to represent the regional model calculations within the mixing time and is subtracted from the regional model output, $C_{RM}$, in order to avoid double-counting emissions. The final concentrations from the coupled model, $C_{coupled}$, are then calculated by adding

the output from an ADMS-Urban run with explicit emissions, also limited to the mixing time ($C_{expl}(\Delta t)$), with the overall expression given as:

$$C_{coupled} = C_{RM} - C_{grid}(\Delta t) + C_{expl}(\Delta t). \tag{1}$$

Additional steps calculate local background concentrations from the regional model which are used as input to the subsequent ADMS-Urban runs, to ensure that the long-range transport and chemical environment is adequately represented for local chemical processes.

The initial implementation of an automated coupled system using ADMS-Urban and the CAMx regional model for Hong Kong is described in Stocker et al. (2014), with evaluation against monitoring data. For the work described in this paper, the coupled ADMS-Urban Regional Model Link (RML) system was further developed to allow the ADMS-Urban runs to be carried out using the ARCHER UK National Supercomputing Service. In the coupled system, meteorology and background concentrations are extracted and used as inputs for separate ADMS-Urban runs for each 5 x 5 km EMEP4UK grid cell, leading to spatially varying meteorology and background concentrations across the modelling domain.

The version of ADMS-Urban (3.4.6) used within the coupled system was slightly older than for the stand-alone runs (4.0.4); the older version was modified for compatibility with the ARCHER supercomputer. There are no relevant differences in terms of dispersion or chemical modelling between ADMS-Urban versions 3.4.6 and 4.0.4.

## 2.5 Emissions Data

### 2.5.1 2012 emissions

EMEP4UK uses anthropogenic emissions of $NO_x$, $NH_3$, $SO_2$, primary $PM_{2.5}$, primary coarse PM ($PM_{2.5-10}$), CO and non-methane VOC. Emissions from the UK are derived from the National Atmospheric Emission Inventory (NAEI, Tsagatakis et al., 2016) for 2012 at 1 km resolution and aggregated to the model's 5 km × 5 km grid. Within Greater London these NAEI emissions are replaced by the emissions prepared for ADMS-Urban as described below. Outside the UK, EMEP4UK uses 2012 anthropogenic emissions provided by the EMEP Centre for Emission Inventories and Projections (CEIP, www.ceip.at/) at 50 km resolution. Shipping emission estimates for seas around the UK are derived from ENTEC (2010), projected to 2012. The anthropogenic emissions are distributed vertically within the model according to their Selected Nomenclature for Atmospheric Pollutants (SNAP) sector, for example road transport emissions (sector 7) are assigned to the lowest layer while power station emissions (sector 1) are assigned to layers between 184 and 1106 m (Simpson et al. 2012, Supplementary material). Biogenic emissions of isoprenes and monoterpene are calculated at each time step according to insolation and surface temperature (Guenther et al., 1995). Emissions of wind-driven sea salt and $NO_x$ from soils are also calculated interactively as described by Simpson et al. (2012), whereas lightning NOx emissions are prescribed. Import of Saharan dust is treated using a monthly dust climatology as a model boundary condition. Resuspension of settled dust by wind is not included.

For ADMS-Urban the emissions for all sources except road traffic for have been taken from the London Atmospheric Emissions Inventory (LAEI) 2010 (GLA, 2013) for the LAEI domain, which covers the area bounded by the M25 orbital motorway. The emissions have been projected from the LAEI base year 2010 to the modelling year 2012. Road traffic emissions have been calculated using activity data from the LAEI. The emission factors used to calculate emission rates are

based on the UK NAEI 2014, which includes speed-emissions data from the COPERT 4 version 10 software tool (Katsis et al., 2012). However, due to uncertainties in $NO_x$ emissions factors for some diesel vehicles and non-exhaust particulate emission factors, adjustments have been made to the published factors to improve consistency with real-world emissions measurements. The adjustments are discussed further in Sect. 2.5.2 and their effects on the modelled concentrations examined in Section 3.1.

The NAEI and LAEI emissions are supplied as a regular, orthogonal 1 km resolution grid in the OSGB coordinate system. The use of the EMEP4UK model in this study requires a conversion to the polar stereographic coordinate system, with re-aggregation onto a grid with a different orientation. This causes some loss of precision in the location of emissions, which is more acute for the ADMS-Urban runs with 1 km gridded emissions than for the EMEP4UK runs with 5 km grid resolution. The average 1 km gridded emissions are reduced by around 5% as a result of the re-gridding process. Within the coupled

modelling system, this reduction of average emissions makes little difference as it only affects the ADMS-Urban run including explicit sources, where concentrations are dominated by the unaffected explicit emissions due to running a limited spatial extent. The change is also small relative to the 'real-world' adjustments and other sources of uncertainty in the emissions. For consistency, the stand-alone local model runs have used the same coordinate system as EMEP4UK and the coupled system in this study.

In ADMS-Urban the road source emissions are modelled with a standard initial mixing height of 2 m above ground, although they may be distributed further upwards due to street canyon effects. Aggregated emissions, represented as gridded volume sources, have a depth of 100 m in the runs without explicit sources, in order to match the behaviour of the EMEP4UK modelling, and a depth of 10 m in the run with explicit sources, where individual point sources are modelled with release heights of 30–200 m.

**2.5.2 Road traffic emissions factor adjustments**

A significant cause of the discrepancies in $NO_x$ and $NO_2$ emission rates between published figures and real-world measurements is the difference in driving conditions between standard test cycles and real journeys, especially those in congested urban traffic (Franco et al., 2013). This issue was highlighted in 2015 when it became apparent that Volkswagen had installed software in their diesel cars that automatically reconfigured the engine during emissions testing (Oldenkamp et

al., 2016). The discrepancies in European vehicle emission rates are expected to begin to decrease due to recent legislative changes (Commission Regulation (EU) 2016/646) which require the use of urban driving cycles and real-world assessment

for emissions testing. Emission factor adjustments are still likely to be necessary for modelling older vehicles which will remain in the active fleet.

Measured volume ratios of $NO_x$ and $NO_2$ to $CO_2$ emissions (a proxy for fuel usage) have been compiled for a range of vehicles, categorised by Euro emission standard and size, with corresponding speeds by Carslaw and Rhys-Tyler (2013).

Measurements were taken at four sites, representing roads in central and outer London. Additional data from bus monitoring campaigns is provided in Carslaw and Priestman (2015) and used for buses running with Compressed Natural Gas fuel. For this study, to make use of these measured data to improve road traffic emissions, the Emissions Inventory Toolkit (EMIT, CERC 2015) software was used to calculate standard ratios of $NO_x$ to $CO_2$ emissions from the raw NAEI dataset for different vehicle types and Euro classes, for average speeds as available in the measured data. 22 vehicle categories were

used for light vehicles and 17 categories for heavy vehicles, with scaling factors calculated from the measured data ranging from 0.80 for Euro II buses to 3.32 for Euro IV buses with Selective Catalytic Reduction (SCR). These scaling factors were used to recalculate $NO_x$ emission rates. This methodology assumes that the standard $CO_2$ emissions factors are substantially more accurate than the $NO_x$ factors, although the former also contain uncertainties. Diesel cars, which make up 41% of the London car fleet (excluding taxis) for 2012 are calculated to have fleet-weighted emissions of $NO_x$ 31% higher due to the

adjustment. Over all road traffic sources in London, the adjustments to emission factors caused an increase in total annual $NO_x$ emissions of 55%. The standard primary fraction of $NO_x$ emitted as $NO_2$ is retained for each vehicle class, but as the $NO_x$ emissions adjustment varies between vehicle classes, the total $NO_2$ emissions do not increase by the same proportion as the $NO_x$ emissions.

Estimates of emission factors used to represent non-exhaust particulate components are relatively unrefined, for example the

20 EMEP/CORINAIR non-exhaust factors use a linear speed-emissions profile and a maximum of ten vehicle categories, in contrast to the hundreds of vehicle categories used for exhaust emissions classification. Analyses of roadside measurements demonstrate that the contribution from brake wear in particular is considerably higher than the published factors (GLA, 2016).

Non-exhaust particulate emission factors were adjusted based on work by Harrison et al. (2012), who analysed

measurements of speciated and size-segregated particulates at the Marylebone Road monitoring site and nearby urban background sites, made during four month-long campaigns between 2007 and 2011. Non-exhaust emissions were found to contribute 77% of the total traffic-related particulate emissions, with 55% of the non-exhaust attributable to brake wear and smaller proportions from resuspension of road dust and tyre wear. Assuming that the standard exhaust emission factors are reliable, the non-exhaust emission factors were scaled in EMIT in order to make up 77% of the total traffic emissions and to

have the correct proportionality between the different components. This is consistent with the approach taken in the LAEI 2013 (GLA, 2016). Applying these adjustments increases the total annual $PM_{10}$ emissions from road traffic sources by 45%. Basing the adjustment of all road non-exhaust emissions on measurements from one site is an approximation, but it is still expected to improve the overall estimates of non-exhaust emissions due to the substantial uncertainty in the standard factors.

The adjusted emissions that reflect real-world conditions as well as possible are hereafter referred to as "real-world" emissions. The total emissions for the LAEI area are summarised by sector in Table 2, including the effects of the adjustments to road transport emissions. Note that CO emissions are unaffected by the road traffic adjustments. A graphical representation of the emissions used in ADMS-Urban is shown in Fig. 4.

### 2.5.3 Time-variation of emissions

In addition to annual average emission rates, it is important for models to capture the temporal variation of emissions in order to represent the short-term variation of concentrations. Within EMEP4UK, the 2012 annual total anthropogenic emissions derived from the inventories are resolved to hourly resolution using prescribed monthly, day-of-week, and diurnal hourly emissions factors (the latter differing between weekdays, Saturdays, and Sundays) for each pollutant and for each of

the SNAP sectors (Simpson et al., 2012).

The stand-alone local model implementation uses an hourly time-varying profile for weekdays, Saturdays and Sundays for all explicit road sources and for aggregated emissions. This time-varying profile is based on a long-term analysis of $NO_x$ measurements in central London (Beevers et al., 2009). The ADMS-Urban runs with gridded emissions within the coupled system use a simplified version of the EMEP4UK monthly and hourly time-varying profiles for $NO_x$ and PM, combined

using a weighting by total emissions for each sector, while runs with explicit emissions use the same profile as in the stand-alone implementation for explicit road sources.

### 2.6 Model evaluation statistics

The following statistics are used to evaluate the modelled concentrations $M$ in relation to the observed concentrations $O$; $n$ is the number of pairs of modelled and observed concentrations, a bar indicates the mean value (e.g. $\bar{M}$), and a subscript

indicates a single parameter value ranked between unity and $n$ (e.g. $M_i$).

Fractional bias ($Fb$) is a measure of the mean difference between the modelled and observed concentrations:

$$Fb = \frac{\bar{M} - \bar{O}}{0.5(\bar{O} + \bar{M})} . \tag{2}$$

Normalised mean square error ($NMSE$) is a measure of the mean difference between matched pairs of modelled and observed concentrations:

$$NMSE = \frac{\overline{(M-O)^2}}{\overline{M}\overline{O}} . \tag{3}$$

Pearson's Correlation coefficient ($R$) is a measure of the extent of a linear relationship between the modelled and observed concentrations:

$$R = \frac{1}{n-1} \sum_{i=1}^{n} \left( \frac{M_i - \bar{M}}{\sigma_M} \right) \left( \frac{O_i - \bar{O}}{\sigma_O} \right), \tag{4}$$

where $\sigma_O$ is the standard deviation of observed concentrations and $\sigma_M$ is the standard deviation of modelled concentrations.

Fraction of modelled hourly concentrations within a factor of two of observations ($Fac2$) is given by the fraction of model predictions that satisfy

$$0.5 \leq \frac{M_i}{O_i} \leq 2.0. \tag{5}$$

The Model Quality Indicator ($MQI$, Thunis and Cuvelier 2016) has been developed through the Forum for air quality modelling in Europe (FAIRMODE, Janssen et al. 2017) as an overall metric of model performance which depends on the measurement uncertainty. The MQI is defined as the ratio between the model bias and twice the measurement uncertainty ($RMS_U$, scaled from the estimated measurement uncertainty at the relevant limit value); lower values reflect better model performance and values of the MQI less than 1 are considered to fulfil the modelling quality objective, in which case model bias is less than twice the measurement uncertainty. This statistic is not defined for $NO_x$ or CO as there are no EU limit values for $NO_x$, whilst CO is typically well below the EU limit value so is not normally assessed. On the assessment target plot, the $MQI$ represents the distance between the origin and a given station point:

$$MQI = \frac{RMSE}{2RMS_U}, \tag{6}$$

where

$$RMSE = \sqrt{\frac{1}{n}\sum_{i=1}^{n}(O_i - M_i)^2} \tag{7}$$

and the ordinate and abscissa correspond to the bias, $(\bar{M} - \bar{O})$, and $CRMSE$:

$$CRMSE = \sqrt{\frac{1}{n}\sum_{i=1}^{n}[(M_i - \bar{M}) - (O_i - \bar{O})]^2}, \tag{8}$$

where both statistics are normalised by twice the measurement uncertainty.

The Robust Highest Concentration ($RHC$) gives an indication of the performance of the model for high hourly concentrations and is defined as:

$$RHC = \chi(j) + \left(\chi - \chi(j)\right)\ln\left(\frac{3j-1}{2}\right), \tag{9}$$

where $j$ is the number of values considered as the upper end of the concentration distribution, $\chi$ is the average of the $j-1$ largest values and $\chi(j)$ is the $j^{th}$ largest value. The value of $n$ is set to 26, as used in Perry et al. (2005).

**3 Results**

Sect. 3.1 assesses the impact of the emissions adjustments on simulated concentrations using the stand-alone ADMS-Urban local model. Sect. 3.2 presents the spatial variation of annual average $NO_2$, $O_3$ and $PM_{2.5}$ concentrations across London predicted by the coupled modelling system while Sect. 3.3 gives detailed evaluation statistics for the regional, local and coupled models based on hourly concentration data for all modelled species. Sect. 3.4 presents additional analysis of the annual average modelled and measured concentrations while Sect. 3.5 concerns the hourly average concentrations and

diurnal cycles for $NO_x$, $NO_2$ and $O_3$. The regulatory standards for $NO_2$, which are defined for annual average and maximum hourly concentrations, have driven this study's focus on these two averaging periods.

## 3.1 Impact of emission adjustments on modelled concentrations

The effect of the adjustment of road traffic $NO_x$ and PM emissions to reflect real-world conditions on all simulated species is shown for background and near-road site types across London in Table 3. This comparison was performed as a preliminary assessment using simulations from the stand-alone ADMS-Urban local model since for this model measured background concentrations are utilized, leading to lower uncertainty in the long-range transport component of concentrations in the stand-alone model than in the coupled system, where the long-range transport contribution is also modelled. The reduced uncertainty means that model errors are the most closely associated with local emissions for this model. The statistics presented are fractional bias (Fb), normalised mean square error (NMSE) and correlation coefficient (R).

The CO concentration results show negligible changes due to the adjustment of emissions, as expected, since CO emissions were not changed. For $NO_x$, $NO_2$, $O_3$ and $PM_{10}$ the emission adjustments result in substantial concentration changes and improvements in Fb and NMSE, especially for near-road sites. For $NO_x$ the concentrations are increased, with Fb values reduced from around -0.3 to close to zero for near-road sites and NMSE reduced substantially; there are smaller concentration and statistics changes for $NO_2$. The change in $NO_x$ concentrations at background sites (+23%) is similar to the change in total emissions (+29%, Table 2), reflecting the direct link from emissions to concentrations for $NO_x$. The change in $NO_2$ concentrations at background sites (+18%) is smaller than both the $NO_2$ emissions change (33%) and the $NO_x$ concentration change, since emitted $NO_2$ contributes only a relatively small amount to total $NO_2$ and due to the time required for chemical processes to convert NO to $NO_2$ which means the response of $NO_2$ concentrations to $NO_x$ emissions is less than linear.

For $O_3$ the impact of the adjusted $NO_x$ and $NO_2$ emissions leads to lower concentrations and reduces the Fb from 0.16 to 0.02 for near-road sites, although there is little change in the NMSE. For $PM_{10}$ concentrations are higher, so the magnitude of the negative Fb values is smaller when using the adjusted emissions, whilst NMSE values are lower over near-road sites but not background sites, which are dominated by regional PM. The large relative contribution of regional PM also causes the concentration changes ($2 - 15\%$) to be substantially smaller than the emissions changes ($25 - 30\%$) for these pollutants. For $PM_{2.5}$ the small over-estimate of concentrations is increased by the emissions adjustment: Fb increases at near-road sites from 0.02 to 0.09. For all species the correlation coefficients remain very similar when using adjusted emission compared to the raw emissions, consistent with the correlation being influenced mainly by the variation in the relative magnitude of concentrations over time, not by their absolute magnitude.

All remaining model results presented in this section use the adjusted road traffic emissions.

### 3.2 Spatial variation of $NO_2$, $O_3$ and PM across London

Annual average contour plots of concentrations for $NO_2$, $O_3$ and $PM_{2.5}$ produced from the hourly coupled regional-to-urban model output using the adjusted emissions data are shown in Figs. 5 to 7. The influence of the M25 London orbital motorway is clearly visible for all three species. The corresponding monitored data are overlaid as coloured points. For $NO_2$,
the highest concentrations (over 100 µg m$^{-3}$ in central London) are found near busy roads, while away from roads the concentrations increase from around 20 µg m$^{-3}$ outside the M25 to around 50 µg m$^{-3}$ in the centre of the urban area. The average $NO_2$ concentration calculated by the coupled model at urban background monitoring sites is 36 µg m$^{-3}$, just below the EU annual average limit value of 40 µg m$^{-3}$ for $NO_2$, while at near-road sites it is 60 µg m$^{-3}$, substantially above the limit value; corresponding measured values are 35 µg m$^{-3}$ for background sites and 61 µg m$^{-3}$ for near-road sites. The modelled
fraction of $NO_x$ which is $NO_2$ increases from 0.43 at near-road sites to 0.59 at urban background sites, due to chemical conversion of emitted NO to $NO_2$. Across London, 333 km$^2$ (13%) of the 2690 km$^2$ urban area within the M25 motorway, excluding road carriageways, exceeds the $NO_2$ annual average limit value, as shown by the yellow, orange and red colours in Fig. 5.

Annual average $O_3$ concentrations show an inverse pattern to $NO_2$, with low concentrations near busy roads ($< 25$ µg m$^{-3}$)
and in the centre of the urban area, due to the effects of titration of $O_3$ by NO. There is no relevant limit value for annual average $O_3$ for comparison. $PM_{2.5}$ concentrations show more uniform background concentrations of less than 10 µg m$^{-3}$ throughout the urban area, with steep increments near roads. The average $PM_{2.5}$ concentration calculated at urban background monitoring sites is 8.9 µg m$^{-3}$ and at near-road sites is 11.2 µg m$^{-3}$, both substantially below the annual average limit value of 25 µg m$^{-3}$ for $PM_{2.5}$. The increase of average concentrations at near-road sites over background sites is similar
to the corresponding measured value, although the overall values are under-predicted (measured 13.7 µg m$^{-3}$ at background sites and 15.7 µg m$^{-3}$ at near-road sites, still well below the limit values). The absolute and relative concentration increment at near-road sites over urban background sites is smaller for $PM_{2.5}$ than for $NO_2$; this difference is captured by the coupled modelling system. A negligible fraction of the urban area (0.003%) exceeds the annual average limit value of 25 µg m$^{-3}$ for $PM_{2.5}$, as shown by the predominantly blue and green colours in Fig. 7. A corresponding plot for $PM_{10}$ concentrations,
showing very similar patterns to $PM_{2.5}$ and negligible exceedences of the annual average limit value of 40 µg m$^{-3}$, is given in Fig. A1.

Plots of annual average concentration of $NO_2$ and $O_3$ against site height, calculated from hourly observations and hourly coupled model output for monitors where both $NO_2$ and $O_3$ are available, are given in Fig. A2. They show generally good agreement between the modelled and observed concentrations, with the increased $NO_2$ and reduced $O_3$ at near-road sites
compared to background sites captured by the model. There is no clear relationship between the concentrations and the site height, especially at the background sites where there is a slightly wider range of site heights.

Overall, the modelled pollutant distributions are closely related to the locations of explicit emissions sources and are also in good agreement with the spatial variation of observed concentrations, especially when viewed at street-scale resolution. The comparisons between modelled and monitored concentrations are discussed in more detail in the following sections.

### 3.3 Evaluation statistics for $NO_2$, $O_3$, CO, $PM_{10}$ and $PM_{2.5}$ for regional, local and coupled models

The performance of the regional EMEP4UK, local ADMS-Urban and coupled models has been assessed using evaluation statistics calculated from hourly concentrations of each pollutant. Table 4 gives statistics for $NO_x$ and $NO_2$ at all of the 42 background and near-road sites at which they are measured, while Table 5 gives statistics for $O_3$, $NO_x$ and $NO_2$ at the 20 sites where $O_3$ is measured, in order to allow detailed analysis of these closely-related pollutants at a consistent set of sites. Table 6 gives corresponding statistics for CO and for the particulate pollutants $PM_{10}$ and $PM_{2.5}$. An additional visual representation of model performance, plots of NMSE against Fractional Bias, is given in Fig. A3. The statistics include those presented for the emissions adjustments in Table 3 (Sect. 3.1) but additionally: the fraction of modelled hourly concentrations within a factor of two of observations (Fac2); and the Model Quality Indicator (MQI). The observed and modelled values of Robust Highest Concentrations (RHC) are also presented. If this value is calculated from all observed or modelled data, it is likely to be dominated by the highest values at a single site, so the approach of averaging individual site values over all sites has been taken in order to calculate more representative values for high observed and modelled concentrations.

The average measured $NO_x$ and $NO_2$ concentrations are lower for sites with $O_3$ measurements (shown in Table 5) compared to all sites (Table 4), as there are a lower proportion of near-road sites with $O_3$. However the general findings are the same for both sets of sites. The regional EMEP model under-estimates $NO_x$ and $NO_2$, as expected for a model using 5 x 5 km gridded emissions. The stand-alone ADMS-Urban model and coupled ADMS-Urban RML system show broadly similar performance for the gaseous pollutants, indicating that the regional model is performing well at simulating the local background gaseous concentrations. For $NO_x$ and $NO_2$, the Fb and NMSE values are much lower when simulated by the stand-alone and coupled models than for the regional model, due to the dominant influence of local emissions in determining concentrations for these short-lived species. Correlation coefficients are higher, with values of around 0.68 for both species for the stand-alone and coupled model simulations, and a similar increase in Fac2. The MQI values for all models except EMEP are less than 1 for $NO_2$, indicating achievement of the FAIRMODE model quality objective. The modelled RHC show both stand-alone and coupled models have good performance in the prediction of peak $NO_2$ concentrations. However these models underestimate peak $NO_x$ values and have values of Fb for $NO_2$ greater than those for $NO_x$, suggesting some over-prediction of $NO_2$ relative to $NO_x$ in general. This is at least in part likely to be due to an over-estimate of the assumed fractions of $NO_x$ emitted as $NO_2$ (Carslaw et al., 2016).

The Fb values for $O_3$ concentrations from the urban and coupled models are also low (0.001-0.02); whilst the NMSE, R values and Fac2 results are fairly similar when comparing all three models to measurements. This reflects the significant contribution of regional background $O_3$ concentrations to the local concentrations within the urban area. The lower values of

MQI for the urban and coupled models show improved overall model performance due to the inclusion of explicit sources. All three modelled RHC values are lower than the observations indicating that although the annual average $O_3$ concentrations are over-estimated, the highest hourly concentrations are under-estimated. This is likely to be due to additional short-term chemistry effects, for instance those caused by large increases in biogenic emissions in hot conditions (Guenther et al., 2006), which are not well captured by the models.

Although no adjustment was applied to the emission rates for CO there is reasonable agreement between model and observations, with particularly good values of Fb and Fac2 for the coupled model. The EU air quality standard for CO is 10 mg m$^{-3}$ for maximum daily 8 hour mean whereas the maximum hourly observed concentration is around 2 mg m$^{-3}$, consistent with no observed exceedences of this standard.

For the particulate pollutants the ADMS-Urban model with measured background concentrations performs markedly better than the coupled model due to poorer performance of the regional model for these pollutants than for the gaseous species in predicting background concentrations. The Fb shows that the regional model underestimates $PM_{10}$ and $PM_{2.5}$ compared to measurements (-0.3 to -0.5); the coupled model also underestimates these species' concentrations compared to measurements but to a lesser extent (-0.1 to -0.4) whereas for the stand-alone model Fb is close to zero. These results reflect the significant regional contribution to local measurements of particulates. Correlation coefficients between modelled and measured concentrations are also higher for the stand-alone model than the other two models, but less so for $PM_{2.5}$ than $PM_{10}$. The modelled PM RHC values for all three models are lower than the observed RHC values, particularly for $PM_{10}$. Very high $PM_{10}$ concentrations are often related to specific local events, such as dust from construction sites, which are not captured by annual average emissions inventories such as the LAEI (Fuller and Green, 2004).

## 3.4 Annual average concentrations for $NO_2$, $O_3$ and $PM_{2.5}$

Some pollutants are closely connected by chemical or physical processes. Here the model performance for $NO_2$ and $O_3$ concentrations is evaluated concurrently. Fig. 8 compares the annual average fractional bias for $NO_2$ and $O_3$ for each model for background and near-road site locations. For many sites the fractional bias of modelled concentrations for both pollutants from each model are within an estimated measurement uncertainty of 15%, as shown on the plot by the square of dotted lines. This is the maximum uncertainty allowable in continuous measurements reported to the EU (EC Directive, 2008). The remaining points, especially those for the regional model at near-road sites, most commonly show that over-estimates of $O_3$ are associated with under-estimates of $NO_2$ while under-estimates of $O_3$ are associated with over-estimates of $NO_2$, as expected from the fast $O_3$ titration chemistry that usually prevails in the urban high $NO_x$ environment. There are two near-road sites where the coupled model over-estimates both $NO_2$ and $O_3$, which does not fit the general pattern.

Assessment target plots (developed as part of the DELTA tool within FAIRMODE, Janssen et al. 2017) allow model performance to be evaluated with an allowance for the measurement uncertainty (Pernigotti et al., 2013), which is particularly relevant for particulate pollutants because of their higher measurement uncertainty compared to gaseous pollutants. Fig. 9 shows the coupled model results for $PM_{10}$ and $PM_{2.5}$ presented on target plots showing the normalised bias

against the centred root mean square error (CRMSE) for each monitoring site, such that the distance of points from the origin gives the value of the MQI and allowance is made for measurement uncertainty. Equivalent plots for $O_3$ and $NO_2$ are given in Fig. A4. The quadrant in which the points are located depends on the magnitude of the relative contributions of any lack of correlation and standard deviation to the model error. Note that the correlation here is calculated with a consideration of measurement uncertainty and is different from the values given in Table 6. The area of the plot with green shading shows where the model errors are within a factor of two of the measurement uncertainty, leading to a value of MQI below 1. All of the $PM_{2.5}$ sites and all except one of the $PM_{10}$ sites lie within the green shading, which indicates achievement of FAIRMODE's model quality objective. The plots show that the errors that occur are mainly associated with negative bias (underestimate), as noted for Fb values in Table 6 (Sect. 3.3), and lack of correlation.

## 3.5 Hourly concentrations and diurnal cycles for $NO_2$ and $O_3$

In this section an evaluation of hourly data and of diurnal cycles is performed. Figures 10, 11 and 12 present frequency scatter plots of hourly $NO_x$, $NO_2$ and $O_3$ concentrations respectively, over all the sites where $O_3$ is measured, split by model and site type. The $NO_x$ plots show a large spread for high concentrations from the regional model at background sites, which is reflected in the coupled model. This may indicate an inaccuracy in the diurnal variation of emissions used in the regional model. The stand-alone local and coupled models capture the large range of observed hourly concentrations, however as expected and noted from the evaluation statistics, the regional model underestimates $NO_x$ at near-road sites.

The scatter of $NO_2$ concentrations is substantially smaller than that for $NO_x$ concentrations, since the dependence of $NO_2$ on $NO_x$ is less than linear on account of the proportion of $NO_x$ that is $NO_2$ increasing with distance from an emission source, due to the chemical reaction of NO with $O_3$, whilst the concentration of $NO_x$ decreases due to mixing and dilution. A high density of points (indicated by the red and orange colours) lying close to the y=x line indicates that a model accurately captures the complex balance between chemical and dispersion processes; Fig. 11 shows that the local and coupled models perform well at both background and roadside sites, but the regional model is unable to represent near-road $NO_2$. For a small number of hours $NO_2$ is over-predicted by the local and coupled models; this may be due to some over-estimate of the fraction of $NO_x$ which is emitted in the form of $NO_2$ (primary $NO_2$), and to limitations in the local chemistry scheme in these cases. The plots for $O_3$ (Fig.12) show generally good agreement between modelled and observed hourly concentrations for all models at all sites but they show an under-prediction of the peak observed values. Some of this may relate to the corresponding over-prediction of $NO_2$, suggesting that the rate of local $O_3$ production through $NO_2$ photolysis is under-estimated. The under-prediction of peak background $O_3$ concentrations by the regional model is reflected in the coupled model results. The under-prediction of peak urban background concentrations by the stand-alone local model using measured rural upwind $O_3$ also indicates an under-estimate of the local generation of $O_3$ through photochemistry within the urban area, for example due to an underestimate of biogenic VOCs in hot conditions (Malkin et al., 2016).

Mean diurnal profiles and 95% confidence intervals for $NO_x$, $NO_2$ and $O_3$ averaged over background and near-road sites are shown in Fig. 13, alongside a specific near-road site (BT4) in order to demonstrate the variability in individual sites. The

BT4 site is located alongside the inner orbital 'North Circular' road with annual average daily traffic of 108,000 vehicles spread across 6 lanes of traffic, and a neighbouring car park. The stand-alone urban and coupled models which include explicit road source emissions (ADMS-Urban and ADMS-Urban RML) typically capture the diurnal cycle of $NO_x$, $NO_2$ and $O_3$ concentrations for the different site types, whilst for the regional model this is only the case for background sites. The diurnal cycle for $NO_2$ strongly reflects $NO_x$ emissions at all site types, showing morning and afternoon traffic-related peaks, but also a dip around midday driven by a peak in $NO_2$ photolysis at this time. $O_3$ peaks around midday but concentrations are lower when $NO_2$ traffic-related peaks occur. Diurnal cycles are similar at both background and near-road sites, although the $NO_x$ and $NO_2$ peak-to-peak concentration ranges are lower and $O_3$ higher at background compared to near-road sites (as noted for the annual average concentrations in Sect. 3.3). The observed diurnal cycle of $NO_x$ concentrations at the BT4 site has a notably higher morning peak than the cycle for the average over all near-road sites; this is less pronounced for $NO_2$ concentrations.

It is apparent that all of the models tend to overestimate $O_3$ when underestimating $NO_2$, especially for near-road sites, as noted for annual-average comparisons in Sect. 3.3. At BT4, the time-variation profile of $NO_x$ is not well captured by the models, but appears closer for $NO_2$ and $O_3$.

## 4 Discussion and Conclusions

This study presents a regional-to-local air quality modelling system which couples the regional EMEP4UK model with the fine scale urban model ADMS-Urban. Model simulations of $NO_x$, $NO_2$, $O_3$, CO, $PM_{10}$ and $PM_{2.5}$ using the coupled system are compared with the regional and urban models run separately and with measurements from background and near-road sites across London for 2012. This choice of base modelling year has allowed detailed assessment of the model chemistry schemes with the ClearFlo summer and winter intensive observation data (Bohnenstengel et al., 2015; Malkin et al., 2016). During the summer of 2012 London hosted the Olympic and Paralympic Games, but no effects from the Games periods were apparent in a comparison of modelled and monitored concentrations at the monitoring site closest to the Olympic park.

The simulations make use of an emissions inventory in which road traffic emissions of $NO_x$, $NO_2$, $PM_{10}$ and $PM_{2.5}$ were adjusted to represent real-world conditions. Using the stand-alone version of ADMS-Urban these were shown to substantially improve both the fractional bias and normalised mean square error, but had little effect on correlations with measured data as these depend on the relative changes in emissions hour by hour which were unaffected by the adjustment.

From the results using the coupled model it is estimated that 13% of the area of London exceeds the EU annual average limit value of 40 µg m$^{-3}$ for $NO_2$ in 2012. This is consistent with the UK report to the EU of the Greater London Urban Area exceeding both the annual average and hourly average limit values (Defra, 2013a). In contrast, concentrations of $PM_{2.5}$ and $PM_{10}$ in London are estimated to have negligible exceedances of the annual average limit values of 25 µg m$^{-3}$ and 40 µg m$^{-3}$ respectively.

The performance of the different modelling approaches used in this study varies depending on the relative importance of regional and local emissions, chemistry and transport processes for different pollutants and site types. The concentrations of the gaseous pollutants $NO_x$, $NO_2$ and CO are dominated by local emissions. This is particularly clear for $NO_x$ and $NO_2$ with large absolute and relative increments in concentration between background and near-road sites. The regional model (EMEP4UK) performs well at background sites but underestimates concentrations of these gases at near-road sites, due to the low resolution of its input emissions data which does not represent individual road sources. The urban (ADMS-Urban) and coupled models both show good agreement compared to measurements at both site types due to the inclusion of explicit source emissions. This means that the coupled model system can be used with confidence for locations or time-periods where rural upwind measurements are not available for use in ADMS-Urban or for assessment of impacts of future emissions or climate change.

The model performance statistics for $NO_2$ are generally better than those for $NO_x$ for all models. This is in part due to the reduced sensitivity to $NO_x$ emissions of $NO_2$ concentrations relative to $NO_x$ concentrations, as exemplified by the analysis of the emissions adjustments. However the clear inverse relationship between model biases for $NO_2$ and $O_3$ is consistent with the local chemistry generally being well modelled, with uncertainty in $NO_2$ and $O_3$ being related to uncertainty in $NO_x$. Comparison between the coupled and stand-alone urban models and measurements of average diurnal profiles of $NO_x$ concentrations suggest the models are capturing the measured features of the profiles, although there is some underestimation in $NO_x$ at roadside around midday.

The concentrations of $PM_{10}$, $PM_{2.5}$ and $O_3$ show more influence from long-range transport than the other gaseous pollutants. Hence the coupled model results for these species are strongly affected by the regional background and the regional model simulation of $PM_{10}$, $PM_{2.5}$ and $O_3$. For the coupled model, simulated $O_3$ agrees well with measured $O_3$ at the background sites, but simulated $PM_{10}$ and $PM_{2.5}$ are largely underestimated compared to background site measurements. However the coupled model still shows a significant improvement compared to the regional model for simulated particulate concentrations, especially at near-road sites, because it includes an explicit representation of local source emissions. The average increment in $PM_{2.5}$ concentrations between background sites and near-road sites is much smaller than for $NO_2$, but is well represented by the coupled system.

The ability of the models to simulate high concentrations has also been investigated. In general the high concentrations are well simulated by the three models and for all pollutants examined in this study except for $PM_{10}$, where the highest concentrations are due to local short-term emission effects, for example construction dust, which is not included in emissions inventories. For $PM_{2.5}$, the urban model gives a reasonable value of the RHC metric and hence of high concentrations, which indicates that these are due to long-range transport, such as from forest fires, which is captured by the measured upwind rural background concentrations but not necessarily by the regional model. The general tendency of the EMEP4UK regional model to underestimate particulate concentrations, both long-term and episodic, has been identified previously (Lin et al.

2017, Vieno 2016b), while all models are affected by local emission effects and uncertainties in measured particulate concentrations (Pernigotti et al., 2013).

Representing the time variation of emissions accurately, including the variation between sites and pollutants, is a challenge for all models and particularly affects the correlation values. In the current work a single time-varying profile was used for all road emissions in the urban and coupled models but the modelling would be improved if more detailed profiles were included. However no time variation data are currently associated with the LAEI. A further model performance evaluation that would be of great interest would be to assess the models against measurements over a wider range of heights, as model predictions are routinely used to calculate building facade concentrations within street canyons. Some very high measurements were carried out during the ClearFlo project, at a height of 180 m on the BT tower, but the authors are not aware of any measurements covering the range of average building heights in central London of 20–40 m (as shown in Fig. 3).

Overall, this study has shown the benefit of coupling a regional atmospheric chemistry transport and dispersion model with a local model in order to calculate detailed spatio-temporal distributions of air pollutants. Such detailed pollutant spatial distributions have applications in health-related exposure analysis (Smith et al., 2016). The coupled system could also be used to assess the effects of air quality policies at a range of scales. An extension to the current study would be to process the hourly model output to assess the exceedence of short-term objectives and combine the results with population data to calculate exposure. The work presented in this paper provides a framework for more detailed examinations of urban atmospheric chemistry, in particular the effects of additional species which interact with $NO_x$, $NO_2$ and $O_3$, and for studies of the effects of the urban heat island and future climate on urban air quality and chemistry.

## 5 Author contributions

IM set up and ran the regional model with support from MV and RD. KJ set up and ran the local stand-alone model with support from JS and CH. CH developed the coupled system with support from JS and DC. IM ran the coupled system with support from CH and RD. KJ and CH processed the model outputs with support from JS. This paper was drafted by CH with input and reviewing by IM, JS, DC, MV and RD.

## 6 Data availability

The authors intend to make the high-resolution hourly output data from the coupled model available on BADC.

## 7 Acknowledgements

This work was funded by the UK NERC under grant NE/M002381/1.

AURN data is © Crown 2017 copyright Defra via uk-air.defra.gov.uk. LAQN data was obtained from the Environmental Research Group of Kings College London (http://www.erg.kcl.ac.uk), using data from the London Air Quality Network (http://www.londonair.org.uk). Both of these datasets are licensed under the terms of the Open Government Licence.

Dr Chun Lin from the University of Edinburgh assisted with the selection of monitoring sites for the model evaluation exercise.

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

**Table 1: Number of monitoring sites per pollutant and site type, with average heights over all sites and by site type. "Background" includes both suburban and urban background sites while "Near-road" includes sites classified as roadside and kerbside. Note that all the selected $O_3$ monitoring sites have co-located $NO_2$ and $NO_x$ measurements.**

| Pollutant | Number of sites | | | Average site inlet height (m) | | |
|---|---|---|---|---|---|---|
| | Total | Background | Near-road | All sites | Background | Near-road |
| $NO_x$ | 42 | 15 | 27 | 3.0 | 3.3 | 2.8 |
| $NO_2$ | 42 | 15 | 27 | 3.0 | 3.3 | 2.8 |
| $O_3$ | 20 | 11 | 9 | 3.3 | 3.6 | 2.8 |
| $PM_{10}$ | 33 | 12 | 21 | 3.1 | 3.5 | 2.9 |
| $PM_{2.5}$ | 11 | 5 | 6 | 3.1 | 3.3 | 2.9 |
| CO | 7 | 3 | 4 | 2.8 | 3.0 | 2.6 |

**Table 2: Total emission rates within the LAEI area by SNAP sector, including the effects of real-world emission adjustments for NO$_x$ and PM from road transport (sector 7)**

| SNAP sector | Description | Total emission rate (Mg y$^{-1}$) | | | | |
|---|---|---|---|---|---|---|
| | | NO$_x$ | NO$_2$ | PM$_{10}$ | PM$_{2.5}$ | CO |
| 1 | Energy production | 7886 | 394 | 307 | 0 | 918 |
| 2 | Domestic and commercial combustion | 3887 | 194 | 99 | 60 | 5428 |
| 3 | Industrial combustion | 4796 | 240 | 192 | 115 | 3367 |
| 4 | Production processes | 948 | 47 | 227 | 22 | 435 |
| 5 | Fossil fuel extraction and distribution | 0 | 0 | 0 | 0 | 0 |
| 6 | Solvent use | 0 | 0 | 28 | 0 | 0 |
| 7 | *Road transport (raw)* | *32147* | *8520* | *2724* | *1420* | *48738* |
| 7 | Road transport (adjusted) | 49673 | 11878 | 3943 | 1916 | 48738 |
| 8 | Other transport | 8439 | 587 | 197 | 150 | 29173 |
| 9 | Waste treatment and disposal | 1647 | 82 | 168 | 150 | 508 |
| 10 | Agriculture | 0 | 0 | 15 | 1 | 0 |
| 11 | Nature | 96 | 5 | 106 | 76 | 0 |
| | *Total with raw road transport* | *59845* | *10070* | *4063* | *1996* | *88568* |
| | Total with adjusted road transport | 77371 | 13429 | 5282 | 2491 | 88568 |
| | Change in total due to adjustments (%) | 29 | 33 | 30 | 25 | 0 |

**Table 3: Model evaluation statistics calculated from hourly average modelled and monitored concentrations for stand-alone ADMS-Urban runs with raw (r) and adjusted (a) road traffic $NO_x$ and PM emissions, and the % change in concentrations due to the emissions adjustment, by site type Bgd (Background) or Nr-Rd (Near-road). Fb – fractional bias in annual average concentration, ideal value 0.0; NMSE – Normalised Mean Square Error in hourly concentrations, ideal value 0.0; R – Correlation coefficient for hourly concentrations, ideal value 1.0.**

| Poll | Site type | Sites | Obs | Mod (r) | Mod (a) | Mod % ((a-r)/r) | Fb (r) | Fb (a) | NMSE (r) | NMSE (a) | R (r) | R (a) |
|------|-----------|-------|-----|---------|---------|-----------------|--------|--------|----------|----------|-------|-------|
| | | | | **Annual mean concentration (µg m⁻³)** | | **Conc. change** | **Model evaluation statistics** | | | | | |
| $NO_x$ | Bgd | 15 | 58.7 | 49.9 | 61.6 | 23.4 | -0.16 | 0.05 | 0.93 | 0.86 | 0.62 | 0.61 |
| $NO_x$ | Nr-Rd | 27 | 149.6 | 105.9 | 149.6 | 41.3 | -0.34 | 0.00 | 0.88 | 0.62 | 0.63 | 0.62 |
| $NO_2$ | Bgd | 15 | 35.4 | 30.4 | 36.0 | 18.4 | -0.15 | 0.02 | 0.32 | 0.27 | 0.66 | 0.67 |
| $NO_2$ | Nr-Rd | 27 | 60.8 | 51.0 | 64.7 | 26.9 | -0.18 | 0.06 | 0.32 | 0.28 | 0.64 | 0.64 |
| $O_3$ | Bgd | 11 | 35.5 | 37.1 | 35.0 | -5.7 | 0.04 | -0.01 | 0.21 | 0.21 | 0.76 | 0.77 |
| $O_3$ | Nr-Rd | 9 | 26.9 | 31.7 | 27.5 | -13.2 | 0.16 | 0.02 | 0.30 | 0.29 | 0.75 | 0.77 |
| $PM_{10}$ | Bgd | 12 | 19.0 | 18.6 | 19.4 | 4.3 | -0.02 | 0.02 | 0.27 | 0.27 | 0.69 | 0.69 |
| $PM_{10}$ | Nr-Rd | 21 | 27.1 | 22.1 | 25.4 | 14.9 | -0.20 | -0.07 | 0.45 | 0.37 | 0.58 | 0.58 |
| $PM_{2.5}$ | Bgd | 5 | 13.7 | 14.1 | 14.4 | 2.1 | 0.03 | 0.05 | 0.29 | 0.29 | 0.76 | 0.77 |
| $PM_{2.5}$ | Nr-Rd | 6 | 15.7 | 16.0 | 17.2 | 7.5 | 0.02 | 0.09 | 0.30 | 0.30 | 0.73 | 0.72 |
| CO | Bgd | 3 | 261.2 | 273.6 | 273.6 | 0.0 | 0.05 | 0.05 | 0.26 | 0.26 | 0.40 | 0.40 |
| CO | Nr-Rd | 4 | 365.4 | 429.0 | 428.9 | 0.0 | 0.16 | 0.16 | 0.41 | 0.41 | 0.48 | 0.48 |

**Table 4: NO$_x$ and NO$_2$ model evaluation statistics calculated at 42 sites for regional (EMEP), local (ADMS-Urban) and Coupled modelled and measured hourly concentrations. Fb – fractional bias in annual average, ideal value 0.0; NMSE – Normalised Mean Square Error in hourly concentrations, ideal value 0.0; R – Correlation coefficient of hourly concentrations, ideal value 1.0; Fac2 – fraction of hourly modelled concentrations within a factor of 2 of observed, ideal value 1.0; MQI – Model Quality Indicator (annual), target value ≤1.0; Average RHC – average over all sites of Robust Highest Concentration calculated for each site (hourly). Note that MQI is not defined for NO$_x$.**

| Poll | Model | Annual mean ($\mu g\ m^{-3}$) | | | Model evaluation statistics | | | | Average RHC ($\mu g\ m^{-3}$) | |
|---|---|---|---|---|---|---|---|---|---|---|
| | | Obs | Mod | Fb | NMSE | R | Fac2 | MQI | Obs | Mod |
| NO$_x$ | EMEP | 117.3 | 50.7 | -0.793 | 2.962 | 0.425 | 0.481 | - | 1111 | 585 |
| NO$_x$ | ADMS-Urban | 117.3 | 118.3 | 0.009 | 0.728 | 0.669 | 0.713 | - | 1111 | 887 |
| NO$_x$ | Coupled | 117.3 | 111.7 | -0.053 | 0.735 | 0.670 | 0.722 | - | 1111 | 750 |
| NO$_2$ | EMEP | 51.8 | 32.7 | -0.453 | 0.819 | 0.459 | 0.639 | 1.31 | 217 | 176 |
| NO$_2$ | ADMS-Urban | 51.8 | 54.5 | 0.051 | 0.293 | 0.688 | 0.829 | 0.93 | 217 | 228 |
| NO$_2$ | Coupled | 51.8 | 51.4 | -0.007 | 0.302 | 0.674 | 0.828 | 0.94 | 217 | 204 |

**Table 5: $O_3$ model evaluation statistics calculated at 20 sites, with statistics for $NO_x$ and $NO_2$ at the same sites, for regional (EMEP), local (ADMS-Urban) and Coupled modelled hourly concentrations. Statistics as defined for Table 4, note that MQI is not defined for $NO_x$.**

| Poll | Model | Annual mean ($\mu g\ m^{-3}$) | | Model evaluation statistics | | | | | Average RHC ($\mu g\ m^{-3}$) | |
|------|-------|------|------|------|------|------|------|------|------|------|
| | | Obs | Mod | Fb | NMSE | R | Fac2 | MQI | Obs | Mod |
| $O_3$ | EMEP | 31.6 | 36.9 | 0.153 | 0.358 | 0.659 | 0.633 | 0.72 | 154 | 130 |
| $O_3$ | ADMS-Urban | 31.6 | 31.7 | 0.001 | 0.241 | 0.777 | 0.664 | 0.37 | 154 | 122 |
| $O_3$ | Coupled | 31.6 | 32.4 | 0.023 | 0.325 | 0.698 | 0.650 | 0.45 | 154 | 129 |
| $NO_x$ | EMEP | 106.1 | 52.5 | -0.676 | 2.865 | 0.401 | 0.555 | - | 1058 | 572 |
| $NO_x$ | ADMS-Urban | 106.1 | 96.6 | -0.094 | 0.787 | 0.709 | 0.728 | - | 1058 | 797 |
| $NO_x$ | Coupled | 106.1 | 97.3 | -0.087 | 0.784 | 0.711 | 0.736 | - | 1058 | 684 |
| $NO_2$ | EMEP | 47.2 | 33.6 | -0.337 | 0.608 | 0.510 | 0.695 | 1.17 | 206 | 172 |
| $NO_2$ | ADMS-Urban | 47.2 | 47.6 | 0.008 | 0.258 | 0.725 | 0.845 | 0.82 | 206 | 204 |
| $NO_2$ | Coupled | 47.2 | 47.4 | 0.004 | 0.262 | 0.721 | 0.845 | 0.88 | 206 | 191 |

**Table 6: CO model evaluation statistics calculated at 7 sites and particulate pollutants statistics calculated at 33 sites (PM$_{10}$) and 11 sites (PM$_{2.5}$) from regional (EMEP), local (ADMS-Urban) and Coupled modelled hourly concentrations. Statistics as defined for Table 4, note that MQI is not defined for CO.**

| Poll | Model | Annual mean (µg m$^{-3}$) | | Model evaluation statistics | | | | | Average RHC (µg m$^{-3}$) | |
|---|---|---|---|---|---|---|---|---|---|---|
| | | Obs | Mod | Fb | NMSE | R | Fac2 | MQI | Obs | Mod |
| **CO** | EMEP | 318.8 | 232.8 | -0.312 | 0.809 | 0.295 | 0.656 | - | 2059 | 1335 |
| **CO** | ADMS-Urban | 318.8 | 359.5 | 0.120 | 0.383 | 0.504 | 0.763 | - | 2059 | 1327 |
| **CO** | Coupled | 318.8 | 317.3 | -0.005 | 0.442 | 0.527 | 0.783 | - | 2059 | 1517 |
| **PM$_{10}$** | EMEP | 24.2 | 17.1 | -0.341 | 0.789 | 0.393 | 0.670 | 0.96 | 205 | 103 |
| **PM$_{10}$** | ADMS-Urban | 24.2 | 23.2 | -0.041 | 0.353 | 0.621 | 0.882 | 0.55 | 205 | 121 |
| **PM$_{10}$** | Coupled | 24.2 | 21.0 | -0.139 | 0.530 | 0.472 | 0.792 | 0.80 | 205 | 110 |
| **PM$_{2.5}$** | EMEP | 14.7 | 8.7 | -0.511 | 0.949 | 0.648 | 0.561 | 0.73 | 110 | 80 |
| **PM$_{2.5}$** | ADMS-Urban | 14.7 | 15.8 | 0.074 | 0.295 | 0.746 | 0.824 | 0.41 | 110 | 98 |
| **PM$_{2.5}$** | Coupled | 14.7 | 10.0 | -0.377 | 0.749 | 0.633 | 0.669 | 0.68 | 110 | 81 |

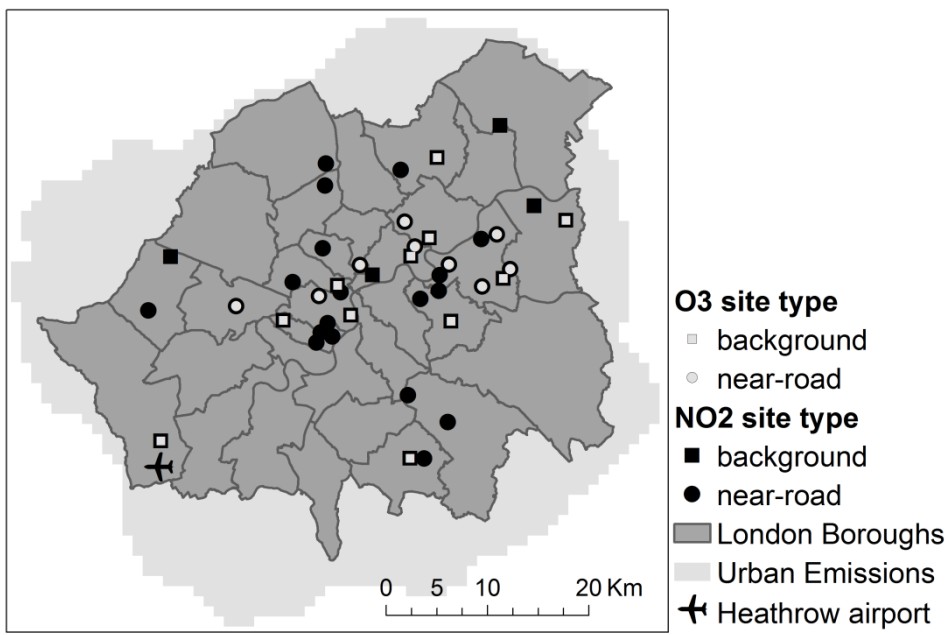

5    **Figure 1: Locations of NO$_2$ (black) and O$_3$ (pale grey) monitoring sites in Greater London, with round symbols for background sites and square symbols for near-road sites. The London borough extents and boundaries are shown for reference, alongside the extent of the locally-modelled emissions and the location of the measured meteorological data at Heathrow airport.**

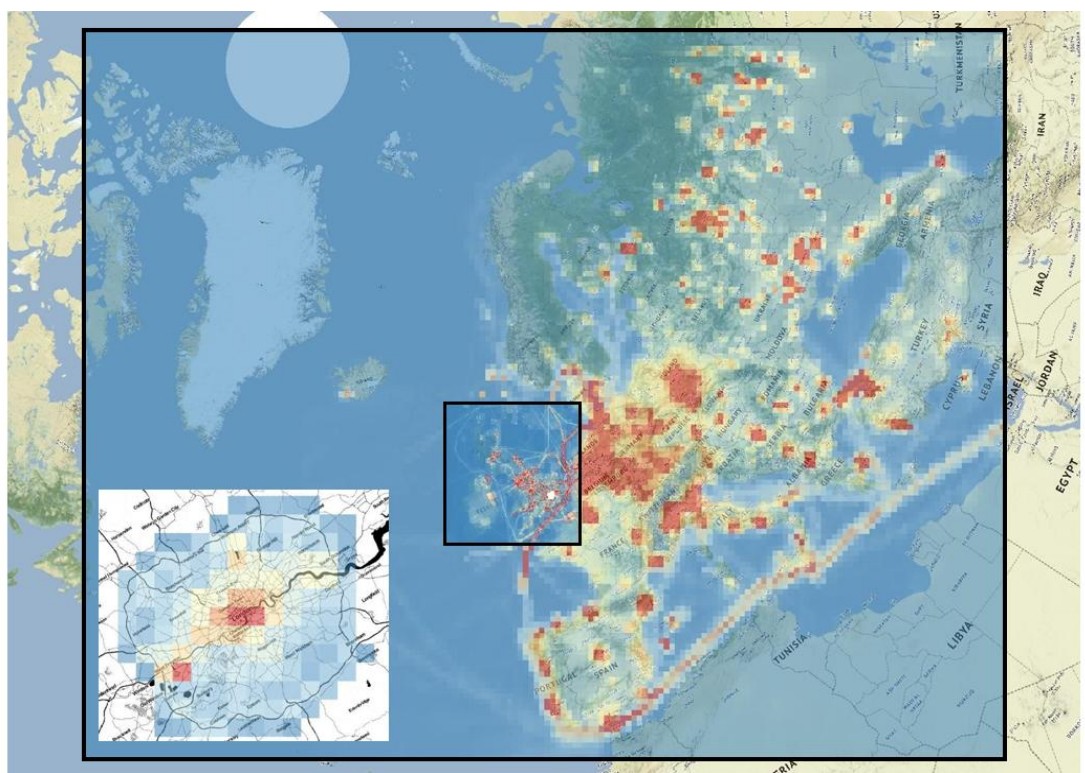

**Figure 2: The nesting structure used by the EMEP4UK model: an inner UK domain simulated at 5 km x 5 km resolution within an outer European domain simulated at 50 km x 50 km resolution, coloured by anthropogenic $NO_x$ emission rates. The greater London region, where EMEP4UK supplies data to the coupled ADMS-Urban RML system, is indicated by white shading on the main figure and is shown inset on a larger scale.**

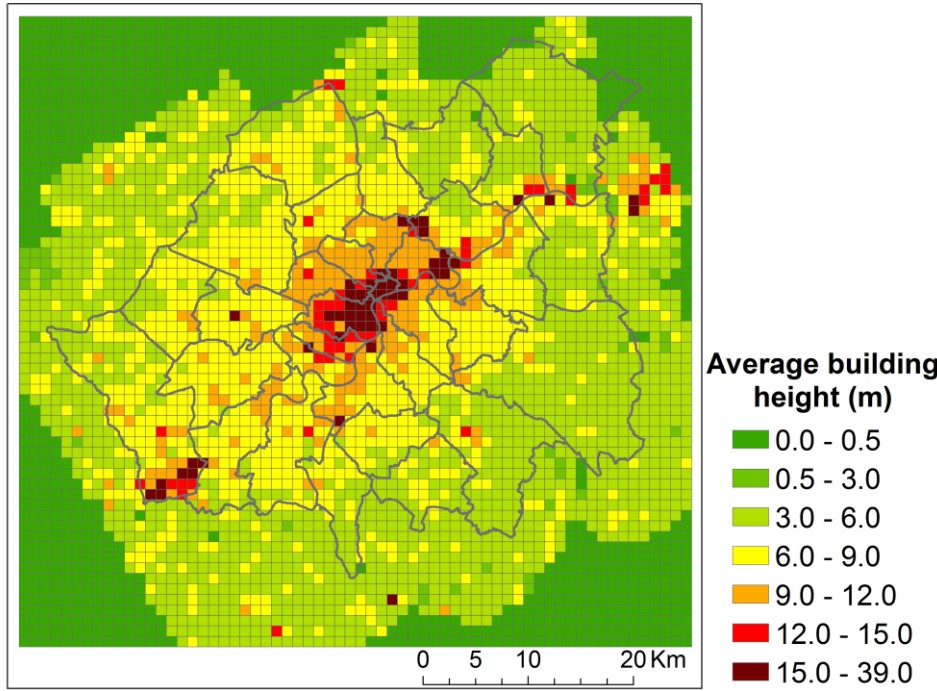

**Figure 3: Spatial variation of the average building height (1 km grid cells) for Greater London.**

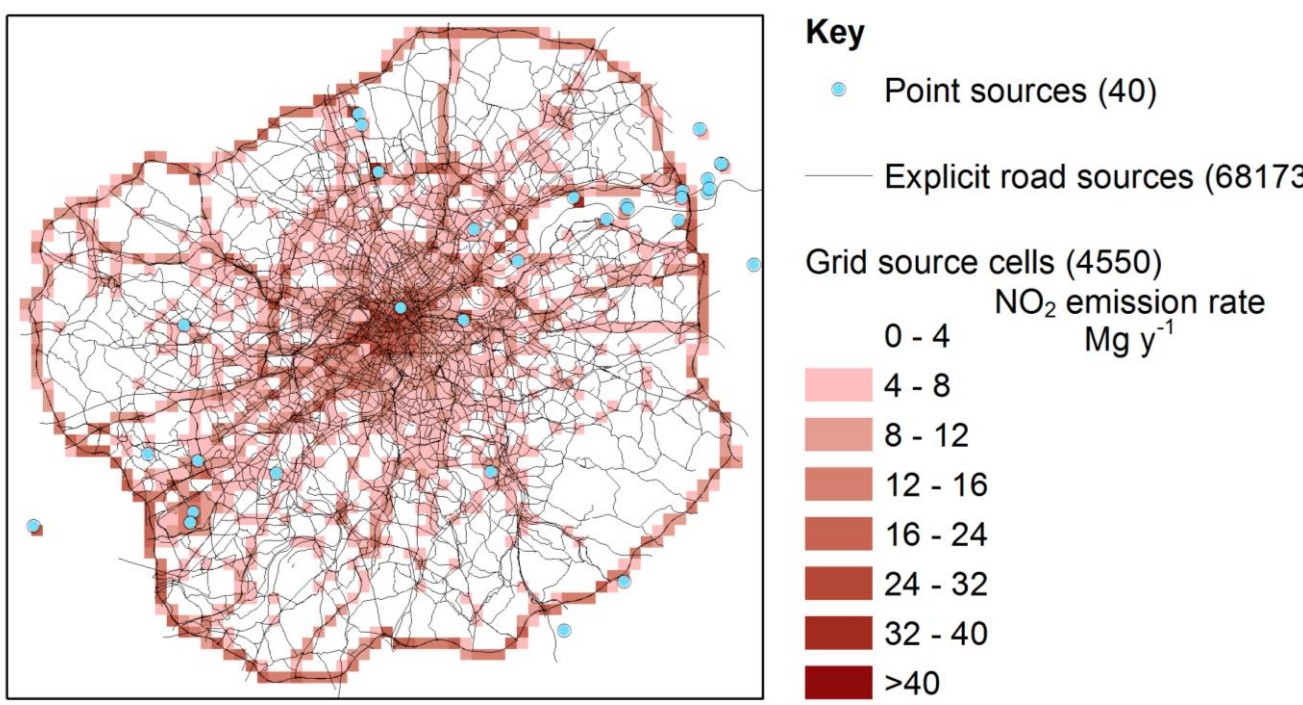

**Figure 4: London 2012 emissions inventory, with source counts given in brackets in the key. Note that railway and river shipping sources are represented by road sources with altered source properties. The gridded sources are 1 x 1 km in extent.**

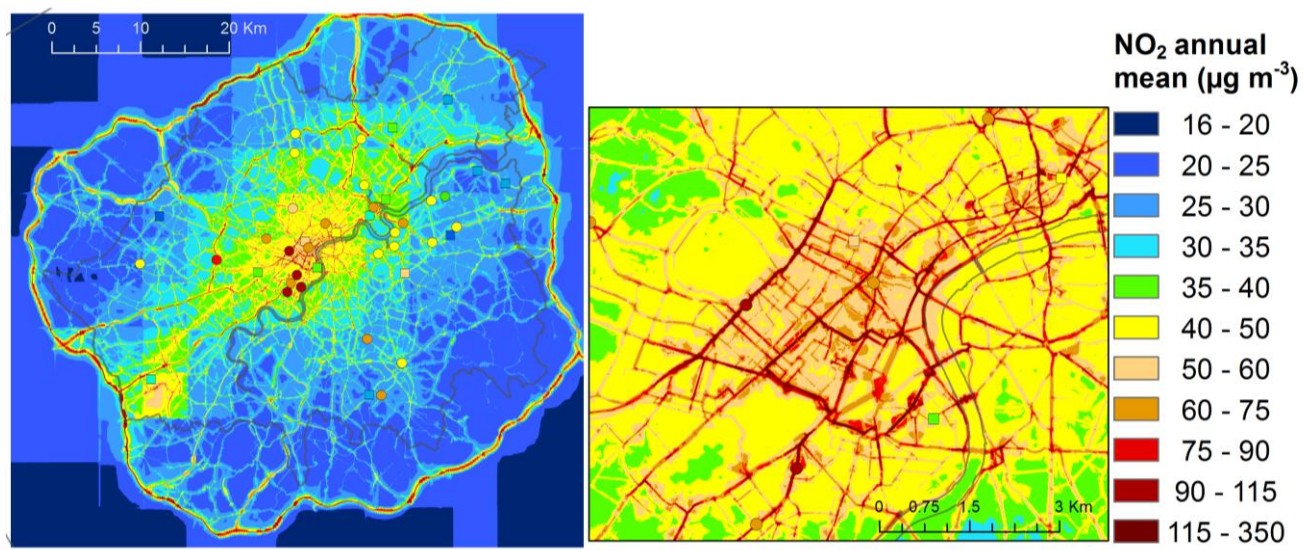

**Figure 5: NO₂ annual average concentrations from the coupled model for the whole of Greater London (left) and an area of Central London (right), with monitoring data overlaid – round symbols for near-road sites, square symbols for background sites.**

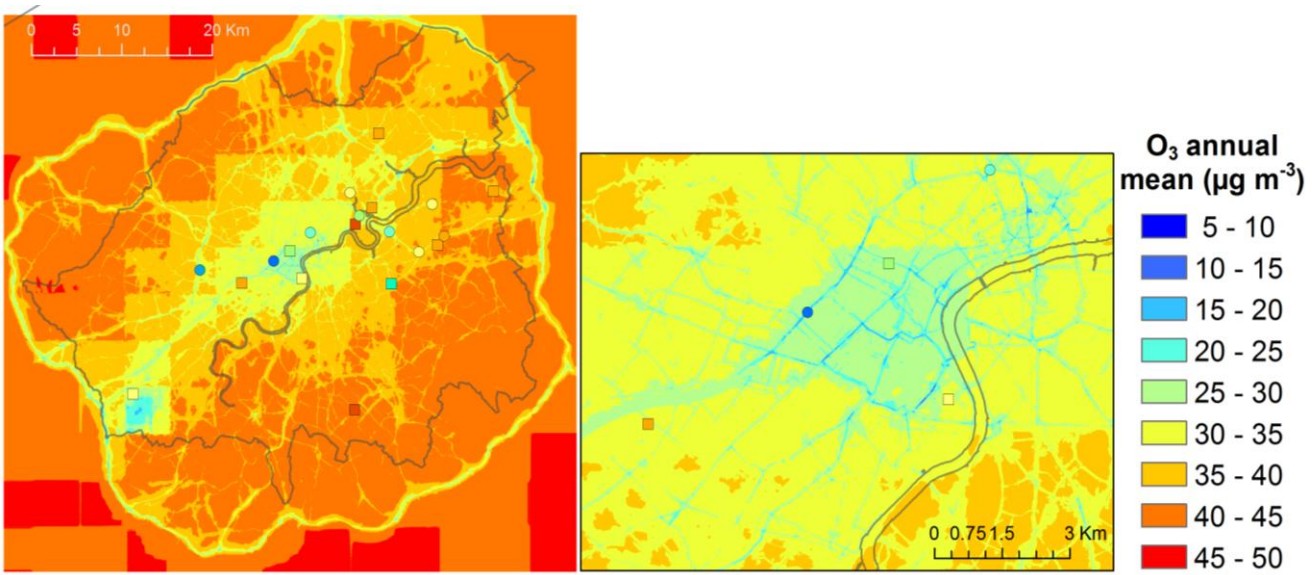

**Figure 6: O₃ annual average concentration contours from the coupled model for the whole of Greater London (left) and an area of Central London (right), with monitoring data overlaid – round symbols for near-road sites, square symbols for background sites.**

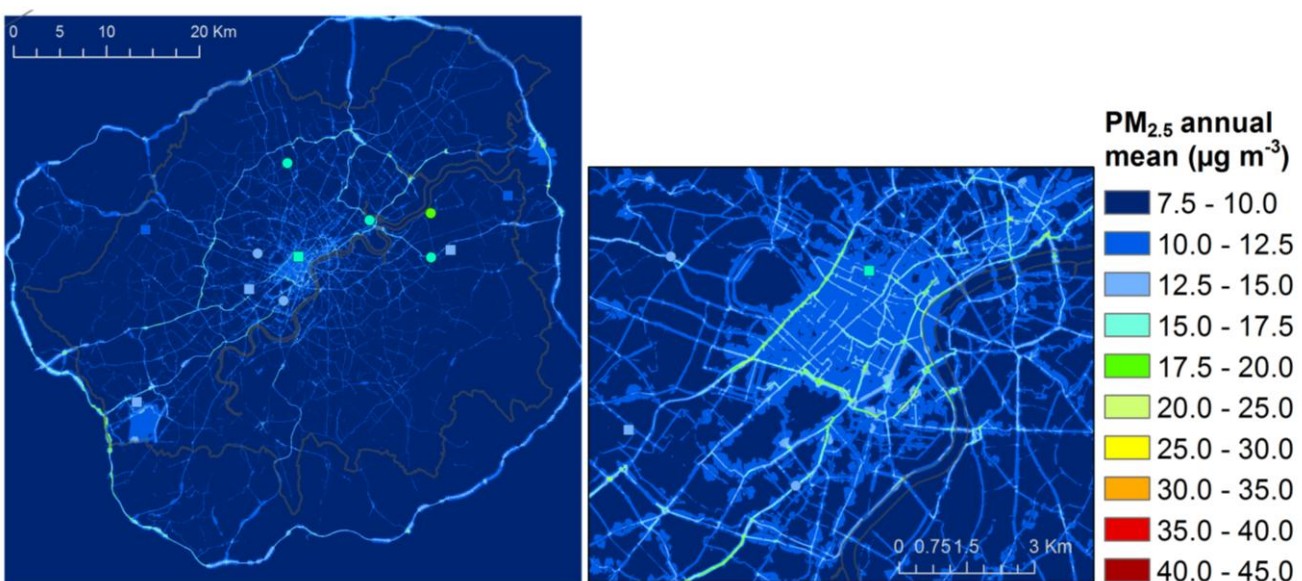

**Figure 7: PM$_{2.5}$ annual average concentration contours from the coupled model for the whole of Greater London (left) and an area of Central London (right), with monitoring data overlaid – round symbols for near-road sites, square symbols for background sites.**

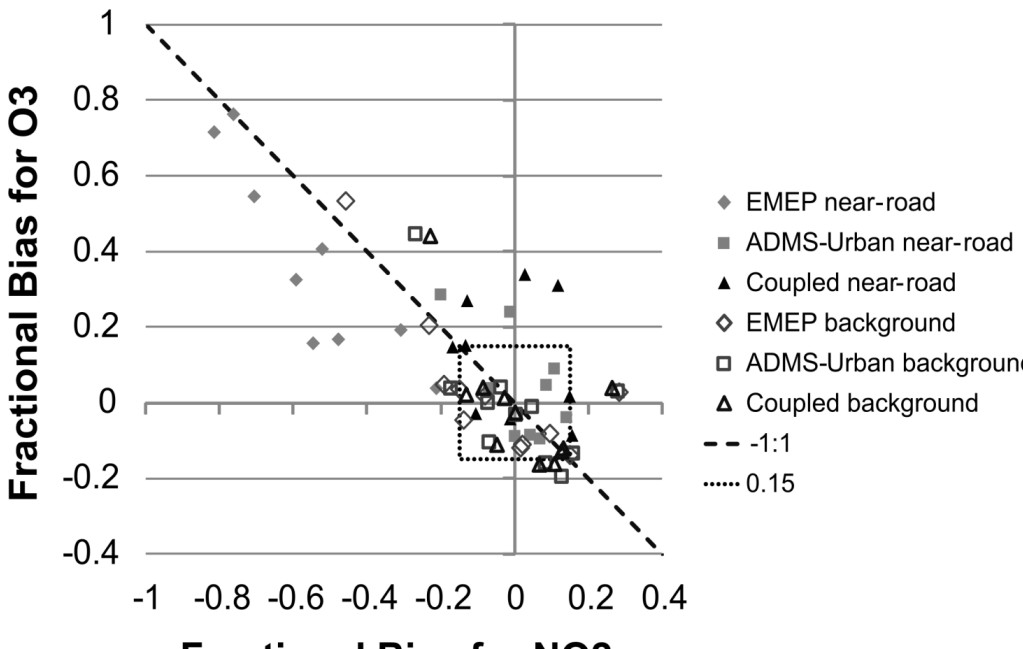

**Figure 8: Scatter plot comparing annual average fractional bias for NO₂ and O₃ for each of the 20 sites where O₃ is measured, for each model. The dotted line represents a fractional bias of 15%, which is the required maximum measurement uncertainty under directive 2008/50/EC.**

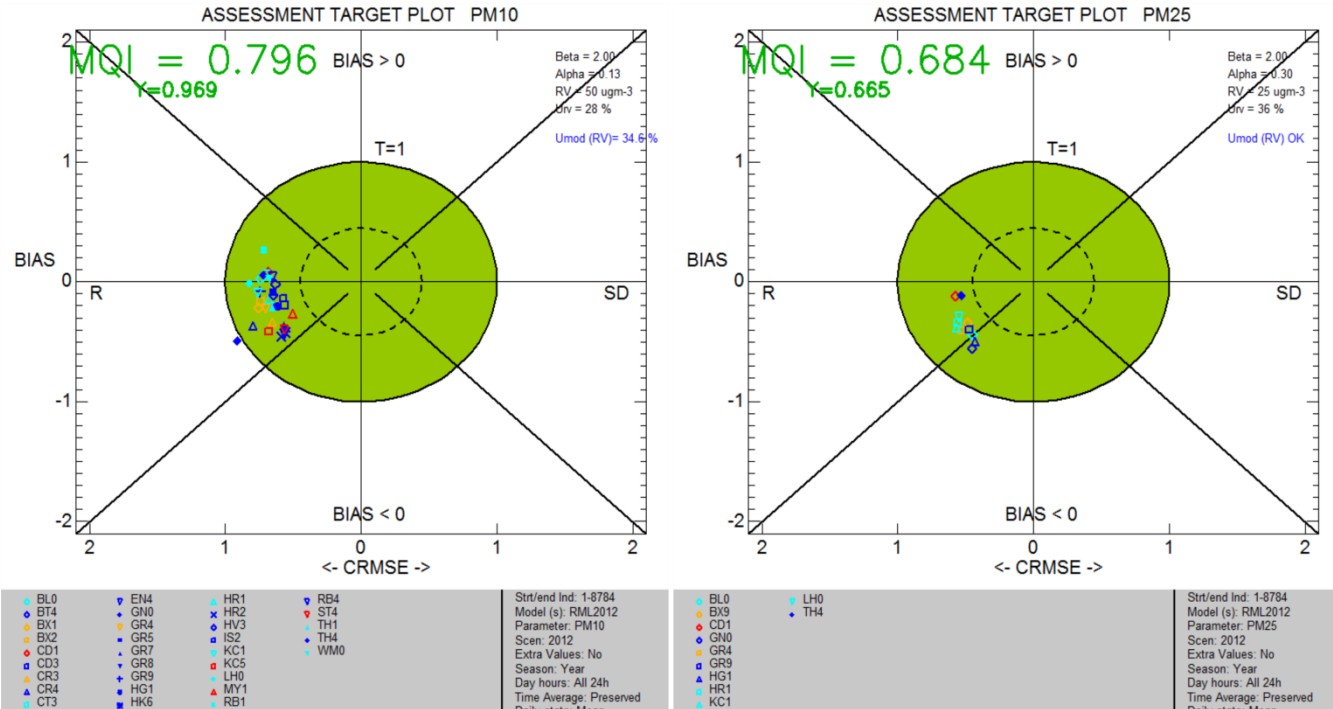

**Figure 9: Model assessment target plots for PM$_{10}$ (left) and PM$_{2.5}$ (right) for coupled system outputs. Each symbol represents a single station and the distance between the origin and the symbol corresponds to the MQI for that station; the ordinate and abscissa correspond to the bias and CMRSE respectively. Good model performance is indicated by points within the green shading.**

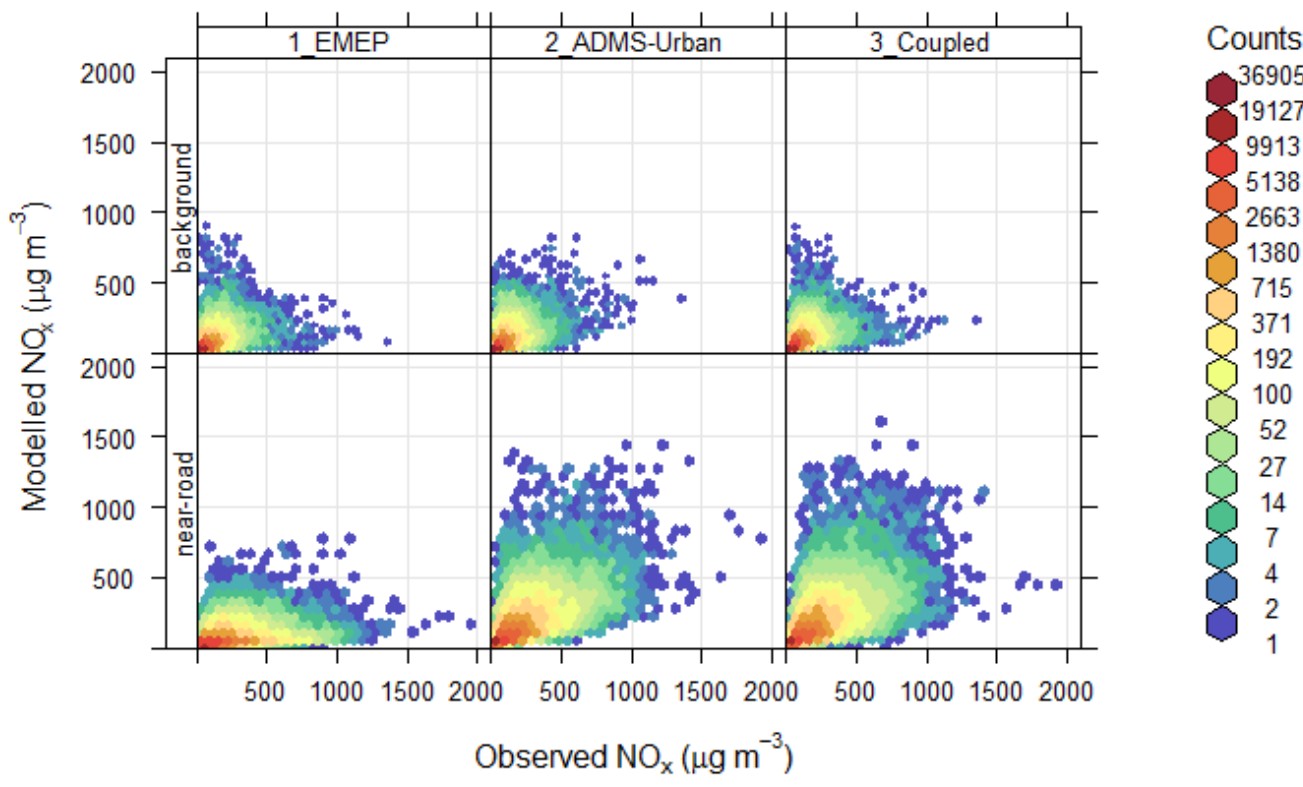

**Figure 10: Frequency scatter plots for each model and site type showing the distributions of hourly average modelled and observed NO$_x$ concentrations (for sites where O$_3$ is also measured), where the colour represents the density of points for a given combination of measured and modelled values.**

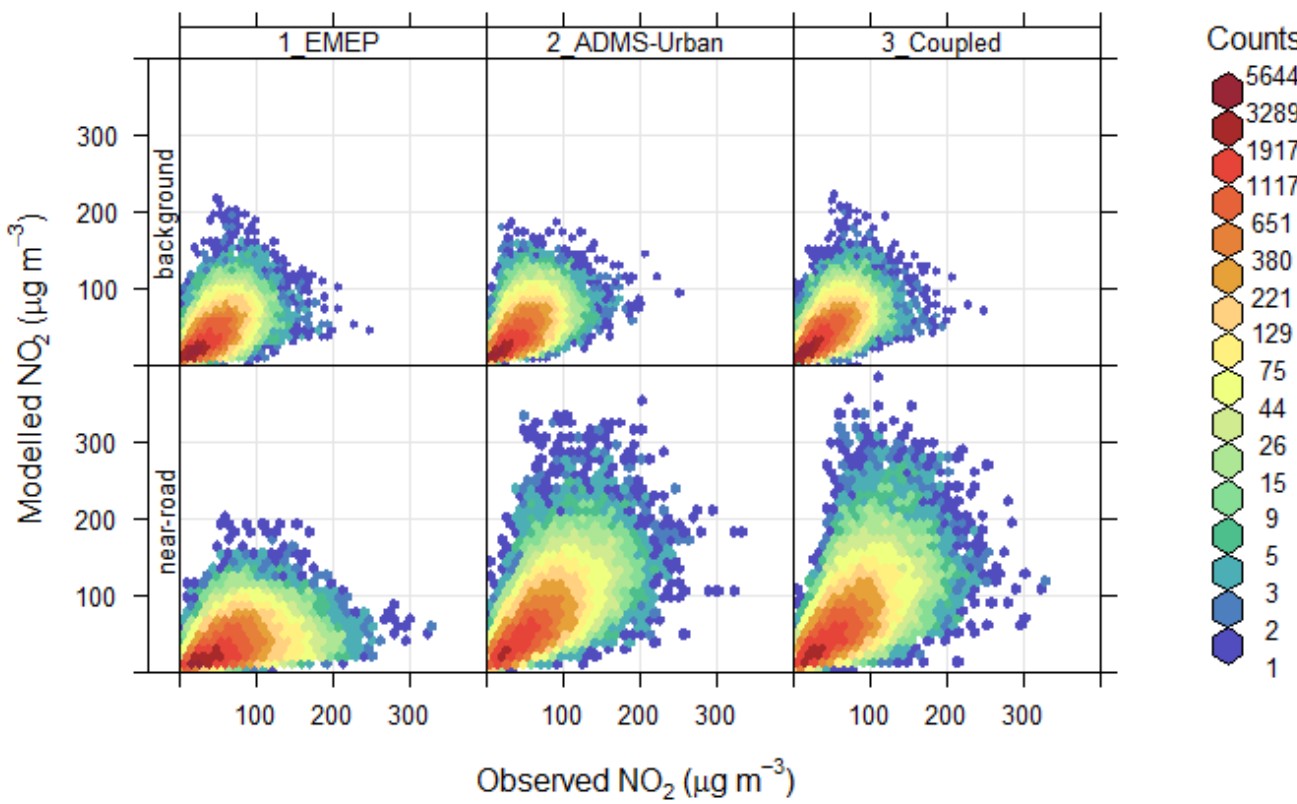

**Figure 11: Frequency scatter plots for each model and site type showing the distributions of hourly average modelled and observed NO₂ concentrations (for sites where O₃ is also measured), where the colour represents the density of points for a given combination of measured and modelled values.**

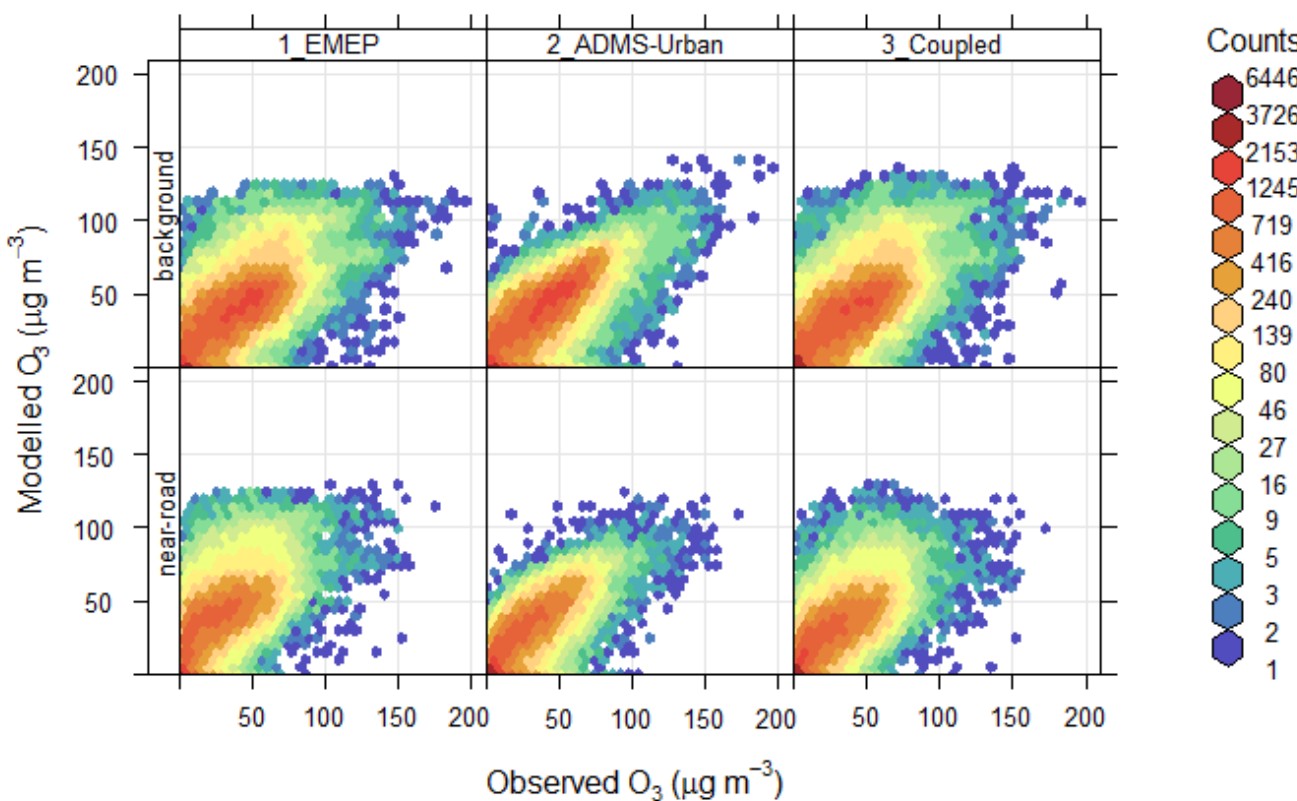

5    **Figure 12: Frequency scatter plots for each model and site type showing the distributions of hourly average modelled and observed O$_3$ concentrations, where the colour represents the density of points for a given combination of measured and modelled values.**

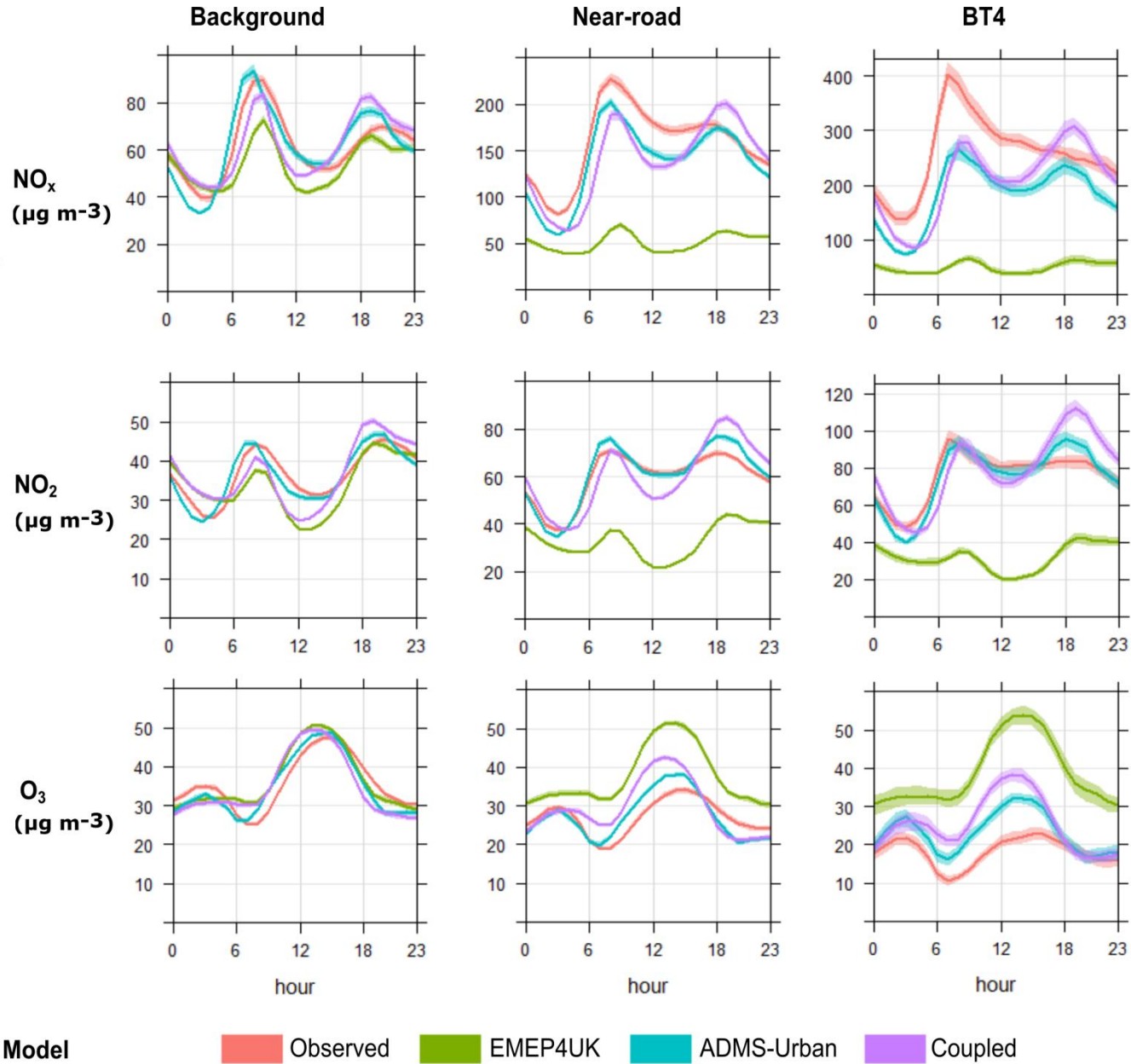

**Figure 13: Diurnal temporal variations of NO$_x$, NO$_2$ and O$_3$ concentrations for the average over all background and near-road sites, and for an individual near-road site, with observations and modelled concentrations from each model. Note different concentration axis limits for each plot. The shaded area around the central line shows the 95% confidence interval in the mean.**

**Appendix A**

Table A1 gives the definitions of the WRF modelling domains used for the EMEP4UK modelling system. The top half of the table gives details of how the WRF domains are defined and the bottom half presents the coordinate system parameters.

Contours of annual average $PM_{10}$ concentrations predicted by the coupled model, overlaid with the observed annual average concentrations, are shown in Fig. A1. The pattern is similar to the plot for $PM_{2.5}$ (Fig. 7), with negligible exceedence of the annual average standard of 40 µg m$^{-3}$, as shown by the mostly blue and green colours. However these model predictions need to be treated with some caution due to the model's general underestimate of $PM_{10}$ concentrations, demonstrated by the statistics given in Table 6.

Plots of annual average concentration against site height for $NO_2$ and $O_3$, calculated from observations and coupled model predictions, are shown in Fig. A2. For background site types, there does not seem to be a clear relationship between concentration and height above ground in the range 2-10 m. The model captures the general differences between near-road and background sites, with increased $NO_2$ and reduced $O_3$.

Following Chang and Hanna (2004), the model performance by site type has been assessed visually as shown in Fig. A3. In general all the models show good performance, with points clustered close to the origin of the graph. The regional model represents background sites adequately but has less good agreement for near-road sites, particularly for $NO_x$ and $NO_2$ where concentrations are dominated by local road emissions, as expected. The urban and coupled models, which represent road sources explicitly, show similar performance for background and near-road sites, with some variation between pollutants. The coupled model shows similar performance to the regional model for background sites, especially for the particulate pollutants and CO, showing the greater influence of the regional model at sites where there are fewer explicit sources represented by the local model.

Figure A4 shows Target plots for $NO_2$ and $O_3$ for the coupled model (both plots only include sites where $O_3$ is measured); results match the measurements to within the target criterion. The model agreement for $O_3$ is particularly good, with many points within the inner circle and a MQI value less than 0.5 indicating that any difference between the modelled and measured values is less than or equal to the estimated measurement uncertainty.

**Table A1: Definitions of WRF modelling domains used for the EMEP4UK modelling system. Note that domain 2 is only used for meteorological modelling.**

| Parameter | Value | | | Comment |
| | Domain 1 | Domain 2 | Domain 3 | |
| --- | --- | --- | --- | --- |
| Parent_id | (1) | 1 | 2 | Nesting hierarchy of domains |
| Parent_grid_ratio | 1 | 5 | 2 | Ratio of grid resolution from parent domain |
| I_parent_start | 1 | 65 | 25 | Cell indices of lower-left cell of |
| J_parent_start | 1 | 40 | 15 | daughter domain in parent domain |
| E_we | 171 | 161 | 221 | Cell extents of each domain |
| E_sn | 134 | 161 | 271 | |
| Geog_data_res | Modis_30s+10m | Modis_30s+10m | Modis_30s+30s | Input land-use data resolution |
| Dx, dy | 50000 | 10000 | 5000 | Grid resolution (m) |
| Map_proj | polar | | | |
| Ref_lat | 57.76 | | | |
| Ref_lon | 6.2041 | | | Coordinate system definition: |
| Truelat1 | 60 | | | applies to all domains |
| Truelat2 | 90 | | | |
| Stand_lon | -32 | | | |

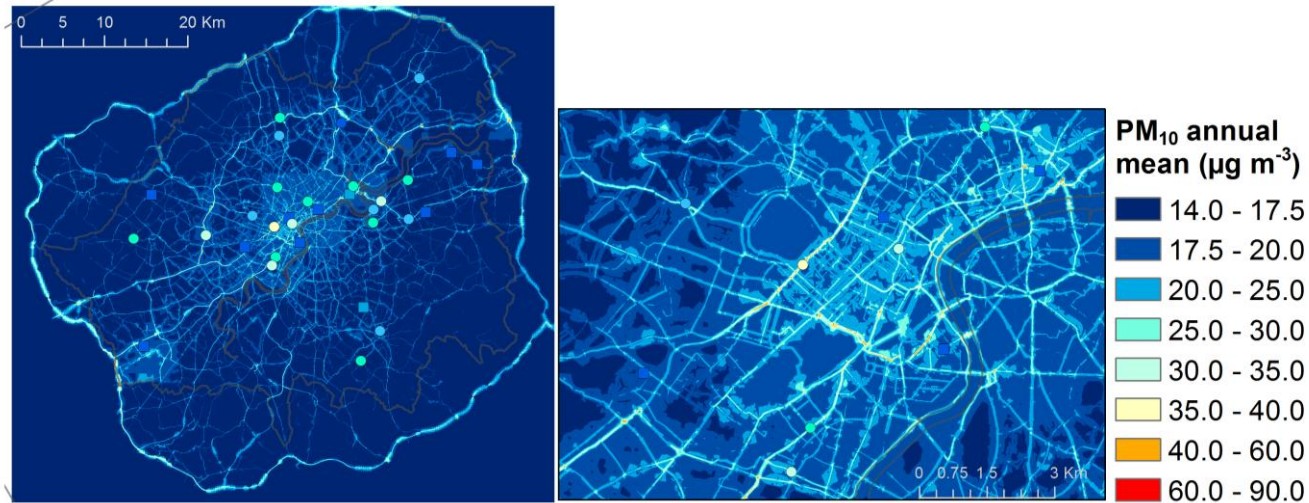

**Figure A1: PM$_{10}$ annual average concentration contours from the coupled model for the whole of Greater London (left) and an area of Central London (right), with monitoring data overlaid – round symbols for near-road sites, square symbols for background sites.**

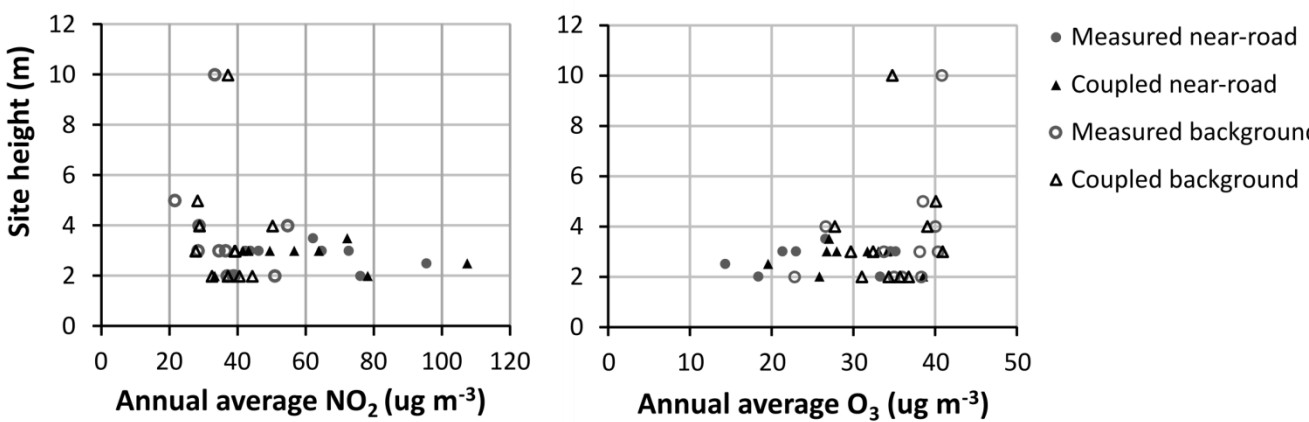

**Figure A2: NO$_2$ (left) and O$_3$ (right) annual average concentrations from measurements and the coupled model, plotted against site height.**

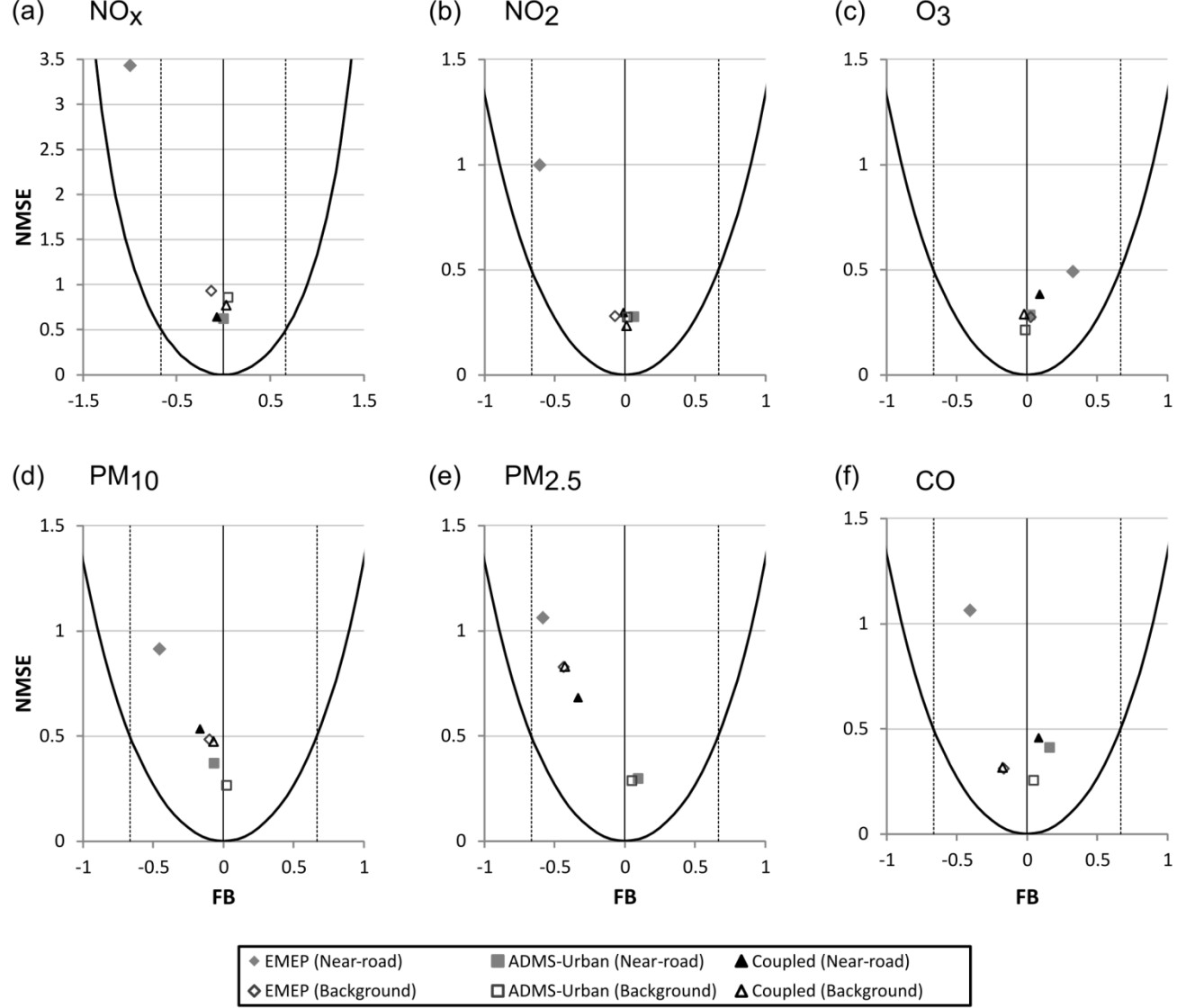

**Figure A3: Model evaluation plots comparing NMSE and Fb for near-road or background sites by pollutant, where improved model performance is shown by the results closest to (0,0). The solid parabola indicates the minimum NMSE for a given Fb, while the dashed lines show modelled results within a factor of two of the observations. Note that the $NO_x$ plot has different axis limits to the other pollutants due to the out-lying EMEP near-road point.**

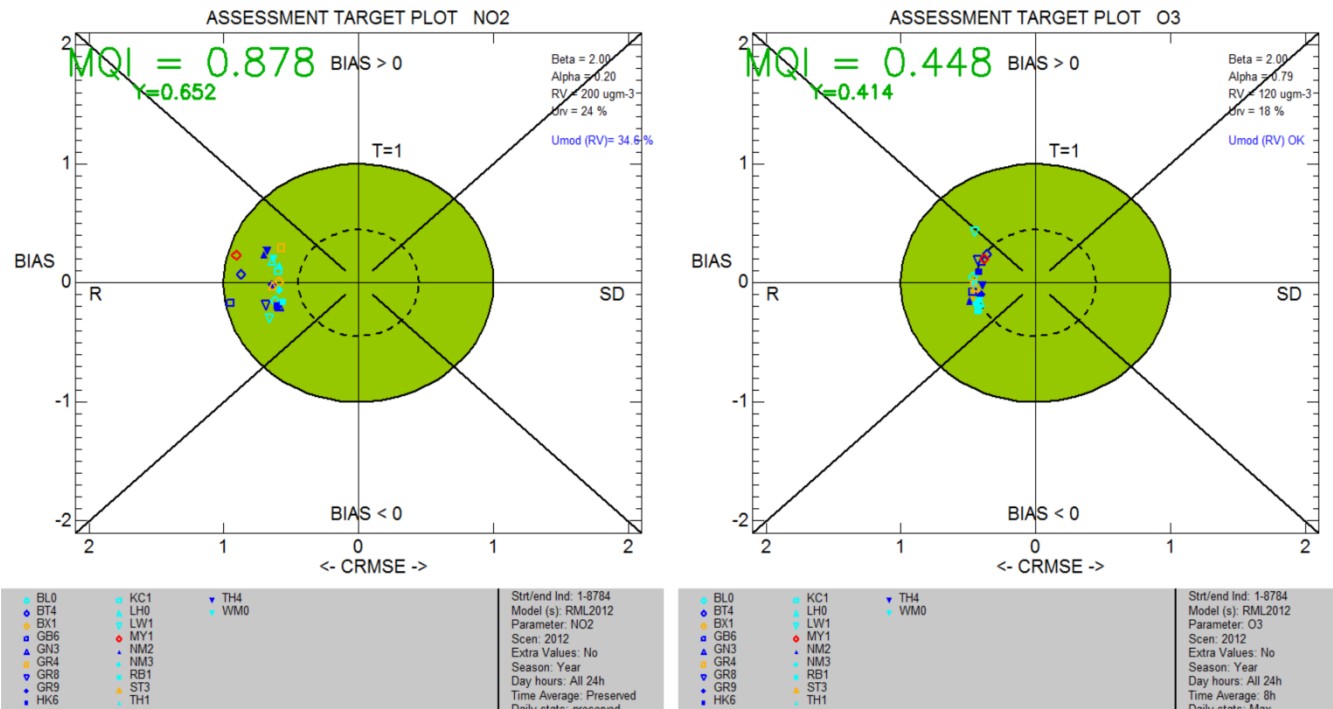

**Figure A4: Model assessment target plots for NO$_2$ (left) and O$_3$ (right) for coupled system outputs. Each symbol represents a single station and the distance between the origin and the symbol corresponds to the MQI for that station; the ordinate and abscissa correspond to the bias and CMRSE respectively. Good model performance is indicated by points within the green shading.**

