# Peer review of "Air quality simulations for London using a coupled regional-to-local modelling system."

_Atmospheric Chemistry and Physics, 2017_

## Referee Comment (RC1) · Anonymous Referee #2 · 28 Feb 2018

Review of Hood et al. "Air quality simulations for London using a coupled regional-to-local modelling system"

The manuscript deals with modelling of air quality in London. The authors present an evaluation of simulations with two different set-ups of a local, "quasi-Gaussian", model and compare the results with a relatively high-resolution Eulerian regional model covering the whole UK. The manuscript also describes a significant but realistic update of the available emissions inventories for London, and use the modelling together with observational data to show that the updated traffic emissions are likely more accurate than the original data.

[Figure]

This is an excellently written paper. The presentation is very clear and pedagogic; the text is supported by a good balance of illustrative and well-drawn figures and a number of information-packed tables. I did not spot a single typo or erroneous formulation. The modelling tools used are clearly state-of-the-art, which yields impressing results.

The manuscript could be published in its current form, or with very minor revision –if the editors deem the scope relevant for ACP. The paper undoubtedly fit in ACP but, in my opinion, would GMD be the optimal choice for this, rather technical, presentation.

General remark: As far as I can see is there no information on the vertical distribution of the emissions. This, I think, is an oversight in an otherwise detailed and complete description of the modelling set-ups. Similarly is there no discussion on what levels the monitoring sites are measuring at. Can there, for example, be systematic differences in the height of the intake between "urban background" (roof-top?) and "kerbside" locations? The possibility that model results and observations are valid at different heights can also be remarked when discussing the models' abilities to reproduce observed concentrations.

Minor issues:

Page 2, Line 14: ". . . up to 4.5." It would be interesting to learn how much of this major discrepancy is due to real-world journeys vs. test cycles in labs, and how much that could be attributed to the so called Volkswagen "diesel-gate" scandal.

Section 2.2: At times it is not clear whether the given information relates to EMEP4UK or the pan-European EMEP MSC-W model. For example:

a) "v4.5" (p4,l1) what model does this refer to?

b) ". . .boundary conditions. . ." (p4,l10-12). This must be the pan-European model. EMEP4UK has the pan-European model on its boundaries, right?

c) ". . .output from . . . WRF. . .". This is EMEP4UK? The pan-European version uses ECMWF? In the description of the one-way nesting of the EMEP models (p4,l1-4) I

take it that the nesting goes directly from a domain with 50 km x 50 km grid-cells to an inner domain with 5 km x 5 km grid-cells. Some scholars would argue that this leap in resolutions is too big. Maybe you should justify or comment this feature of EMEP4UK.

Section 2.3 (p5,l8): Why do you mention that ADMS-Urban has "...two street canyon formulations"? It is not clear from the presentation if one or both are used in the present simulations.

(p5, line 25): Maybe you could indicate the location of "Heathrow" in Fig. 3 or Fig 1. Would be helpful for readers not familiar with the London geography.

Section 2.4 (p6,l4-...): The paragraph on how to avoid counting the emissions twice is difficult to follow. Maybe this paragraph can be re-formulated?

Section 2.5: A description of the vertical release height is missing. It is also stated that the horizontal re-gridding reduces the total emissions by ca. 5%. It is not clear whether it is left this way or if this missing 5% is "put back" to the regridded emissions to not loose such a significant amount of the emissions.

Section 3.1 (p9,l20-21): Consider defining, mathematically, fractional bias (Fb), normalised mean square error (NMSE) and correlation coefficient (R).

Section 3.2: a)The coloured points in Figs 5-7 (and Fig. A1) are sometimes difficult to discern, maybe consider adding a black or white frame.

b) The legend of Figs. 5-7 (and Fig. A1) should probably read "Modelled ... annual average...".

Section 3.3 (p.11,l8-9): Consider defining, mathematically, the MQI.

(p11,l13-14): Eq. (1) is not complete and should start with: RHC=X(n)+ ...

(p12,l19): "...Fb~0." should probably read "...Fb is close to 0..." or similar. Harmonize spelling of "...standalone..." (p12, l19), "...stand-alone..." (p12,l21).

Section 3.4: (p13,l6) "…normalised bias…" is that the same as "fractional bias" which is used elsewhere in the presentation?

(p13, l6-7)? What is "centred root mean square error (CRMSE)"? Consider write out the mathematical definition.

(p13,l10) Unclear to me how "…the correlation here is calculated with a consideration of measurement uncertainty…"; I assume it is described in the DELTA tool mentioned earlier in the paragraph.

Section 3.5 (p13, l17-20): I note that you only devote four lines of text to 2 comprehensive and illustrative figures. If the editor deems your presentation to long you may omit this paragraph along with Figs. 10 and 11.

Section 4: (p14, l8). Why did you choose to model the year 2012 (London Olympics?), I understand the underlying LAEI emissions inventory is valid for 2010. It may have been more interesting with a more recent year (or two contrasting years!).

---

## Referee Comment (RC2) · Anonymous Referee #1 · 25 Apr 2018

General comments:

The paper addresses relevant scientific questions within the scope of ACP. The authors do not clearly support to present novel concepts, ideas, tools, and data, but they are applying integrated model approaches to investigate regional and local AQ. A thorough comparison among the approaches is given. The overall presentation is well-structured and clear; some insertions are proposed below. Some typos and grammar issues noticed should be corrected after the final revision by the authors.

Specific comments:

Abstract: General introductory phrases should be avoided. Instead, more results

should be present. Introduction: A more extensive review (and references) is proposed with respect to the regional and urban models (p. 2 line 20 and on).

P2. Lines 29-31: The use of boundary concentrations from measurements for model applications has limited applicability not only for future applications but also for diagnostic runs due to several reasons: temporal analysis of measurement data (if not hourly), and mostly, adequate spatial information in the area of interest.

Sect. 2.2: The wide use of EMEP model should be supported by references (p3, line 28). Which is the simulation period of this model? Please provide the simulation domain using coordinates, as well (mainly for reproduction purposes).

Lines 3-4: The 1st vertical layer seems quite deep. Where do you base such a choice? Can you support this height for your region?

Line 8: please provide reference(s) for the VBS scheme you use.

Lines 5-12: which is the size distribution of aerosol species within the regional model?

Sect. 2.3: Some important information for the city scale model is not given. i.e. which are the simulated pollutants (chemistry, sizes in case particles)? Which spatial criterion (coordinates) defines the simulation domain? Which is the simulation period of this model? which is the temporal resolution of the simulations/outputs?

The way/method used to provide ic/bc conditions to the city scale model is not described (pollutants, spatial distribution of used stations, temporal resolution etc).

Sect. 2.4: the coupling method is quite unclear. Despite based on a previous study, it is better briefly described shortly here as well.

Lines 17-18: unclear. Please rephrase

*it may be helpful to provide a table with the model applications used (incl. emission scenarios), for each purpose.

Sect. 3: the justification with respect to model errors is not quite understood (lines 16-21)

Sect 3.2: not knowing the simulation period, it is a question how the annual average is calculated (e.g. from representative simulation events?)

Sect. 4. I would rather prefer a balance between evaluation results and AQ results for the area of interest. At the moment, mostly the model performance is discussed rather than AQ issues.

Tables are quite many in number. Consider either merging those with similar information or moving the less important to the Appendix.

Some issues on the figures to be corrected:

Figure2: the quality of the figure seems poor.

Figure 4: the mass emission rates are not pronounced on the map.

Figures 5-7: the information on the model (system) used to produce these outputs is omitted.

Figure 9: the figure is poorly described in the legend. Bear in mind that each figure should stand alone.

Figures 10- 11: the axis numbers of the middle plots are on top.

Figure 12: axis titles are duplicated.
* * *

---

## Author Comment (AC1) · 15 Jun 2018

**Referee comment 1 (Referee 2)**

| Referee comment | Author's Response | Changes in manuscript |
|---|---|---|
| General remarks: missing information about vertical distribution of emissions and vertical heights of monitoring sites, in particular whether there is a systematic difference between near-road and background locations and whether this affects the validity of model outputs. | Background sites have average heights less than 1m higher than the average for near-road sites. There is no clear relationship between the measured concentrations and site heights. | Table 1 added average heights by site type. Section 2.1 sentence added about monitoring heights by site type. Additional information added about vertical distribution of emissions (refer to separate comment for details). New appendix plot of annual average concentration with site heights for $NO_2$ and $O_3$. Related text added to section 3.2. |
| P2 L14: How much discrepancy is due to real-world journeys vs. Test cycles; how much to Volkswagen 'diesel-gate'? | The measurements used for the real-world adjustment were made in 2012, so may include some affected Volkswagen vehicles, but the proportion of those vehicles in the measured fleet is unknown. | Text added to section 2.5.2. |
| Section 2.2: more clarity needed to differentiate EMEP4UK and EMEP MSC-W
 a) V4.5 (P4 L1) – which model?
 b) 'boundary conditions' – for European EMEP?
 c) 'output from WRF' – for EMEP4UK? Default in EMEP is ECMWF?
 Add a comment on nesting directly from 50x50 km EMEP to 5x5km EMEP4UK. | | a) Version no. clarified as applying to EMEP
 b) The boundary conditions have been clarified as applying to the European domain.
 c) WRF output is used for both the European and UK domains in EMEP4UK.

 An intermediate 10 x 10 km resolution domain is used for WRF to ensure numerical stability, but is not required for the chemistry transport calculations (Vieno et al. 2010). Table A1 has been added with full WRF domain definitions. |
| Section 2.3 (P5 L8): Why do you mention that ADMS-Urban has "...two street canyon formulations"? It is not clear from the presentation if one or both are used in the present simulations. | | The description of the basic canyon model has been removed and a statement added that advanced street canyon option has been used for all roads with adjacent buildings. |

| Referee comment | Author's Response | Changes in manuscript |
|---|---|---|
| P5 L25: Maybe you could indicate the location of "Heathrow" in Fig. 3 or Fig 1. Would be helpful for readers not familiar with the London geography. | | Aeroplane symbol added to Fig. 1 to show location of Heathrow airport. |
| Section 2.4 (P6 L4-...): The paragraph on how to avoid counting the emissions twice is difficult to follow. Maybe this paragraph can be re-formulated? | | The paragraph describing the concept of the coupled model to avoid double-counting emissions has been revised and expanded. |
| Section 2.5: a) A description of the vertical release height is missing. b) It is also stated that the horizontal re-gridding reduces the total emissions by ca. 5%. It is not clear whether it is left this way or if this missing 5% is "put back" to the regridded emissions to not lose such a significant amount of the emissions. | b) In the coupled system the 5% re-gridding reduction of 1x1 km emissions only affects the run with explicit emissions, where the unaffected explicit emissions dominate the concentrations. In addition within this system only 25 1 km cells are modelled at once and the wider urban area effects are represented by 5x5 km emissions in the regional model, which are affected to a lesser extent. In general this 5% change is considered small relative to the real-world emissions adjustment and general emissions uncertainties. | a) More information about EMEP vertical profiles given in first paragraph of section 2.5.1. Final paragraph extended to cover vertical structure of emissions used in ADMS-Urban. b) Additional sentences explaining why the change in emissions due to re-gridding is unimportant have been added to the final paragraph of section 2.5.1. |
| Section 3.1 (P9 L20-21): Consider defining, mathematically, fractional bias (Fb), normalised mean square error (NMSE) and correlation coefficient (R). | | New section 2.6 added with all evaluation statistic definitions. |
| Section 3.2: a) The coloured points in Figs 5-7 (and Fig. A1) are sometimes difficult to discern, maybe consider adding a black or white frame. b) The legend of Figs. 5-7 (and Fig. A1) should probably read "Modelled ... annual average...". | a) A thicker border on the monitoring site points could hinder understanding of model-monitor matching at the local scale. | b) Captions modified, also taking into account comments from Referee 1. |
| Section 3.3 (P11 L8-9): Consider defining, mathematically, the MQI. | | New section 2.6 added with all evaluation statistic definitions. |
| P11 L13-14: Eq. (1) is not complete and should start with: RHC=X(n)+ ... | | Equation completed by adding 'RHC=' and moved to new section 2.6 |

| Referee comment | Author's Response | Changes in manuscript |
|---|---|---|
| P12 L19): "...Fb~0." should probably read "...Fb is close to 0..." or similar. | | Text altered. |
| Harmonize spelling of "...standalone..." (p12, L19), "...stand-alone..." (P12, L21). | | Harmonised as stand-alone throughout text, as this was the more commonly used variant. |
| Section 3.4: (P13 L6) "...normalised bias..." is that the same as "fractional bias" which is used elsewhere in the presentation? | No, the normalised bias is normalised by the mean of the observations, whereas the fractional bias is normalised by the average of the mean of the observations and the modelled values. | Defined in new section 2.6. |
| P13 L6-7: What is "centred root mean square error (CRMSE)"? Consider write out the mathematical definition. | | Defined in new section 2.6. |
| P13 L10: Unclear to me how "...the correlation here is calculated with a consideration of measurement uncertainty..."; I assume it is described in the DELTA tool mentioned earlier in the paragraph. | | Defined in new section 2.6 |
| Section 3.5 (P13 L17-20): I note that you only devote four lines of text to 2 comprehensive and illustrative figures. If the editor deems your presentation too long you may omit this paragraph along with Figs. 10 and 11. | A corresponding plot for NOx has been added and the discussion extended. | A new figure 10 has been inserted and the first paragraph of section 3.5 significantly expanded. |
| Section 4: (P14 L8). Why did you choose to model the year 2012 (London Olympics?), I understand the underlying LAEI emissions inventory is valid for 2010. It may have been more interesting with a more recent year (or two contrasting years!). | The project under which this modelling was undertaken began in 2014. One of its aims was to use the ClearFlo monitoring campaign data from summer and winter 2012 to investigate the regional and local model chemistry schemes, which led to this choice of base modelling year. The 2012 modelling and monitoring data for sites nearest the Olympic park has been examined and no effects from the Olympic and Paralympic periods were apparent. | An explanation of the choice of base year has been added to the first paragraph of section 4, with reference to assessment of chemistry schemes against ClearFlo data. |

**Referee comment 2 (Referee 1)**

| Referee comment | Author's Response | Changes in manuscript |
|---|---|---|
| Abstract: General introductory phrases should be avoided. Instead, more results should be present. | | The abstract has been edited and extended to make it more focused on the results. |
| Introduction: A more extensive review (and references) is proposed with respect to the regional and urban models (P2 L20 and on). | | More regional and local models have been mentioned in the introduction. |
| P2 L29-31: The use of boundary concentrations from measurements for model applications has limited applicability not only for future applications but also for diagnostic runs due to several reasons: temporal analysis of measurement data (if not hourly), and mostly, adequate spatial information in the area of interest. | | Additional sentence inserted relating to use of measured upwind concentrations. |
| Sect. 2.2: a) The wide use of EMEP model should be supported by references (P3 L28).
b) Which is the simulation period of this model?
c) Please provide the simulation domain using coordinates, as well (mainly for reproduction purposes). | c) The European domain is a standard EMEP configuration. The full WRF domain definitions, including the coordinate system definition, have been included as table A1. | a) The second and third sentences of section 2.2 have been combined to make the references clearer.
b) Hourly output resolution for 2012 stated in added final sentence of section 2.2.
c) Appendix Table A1 added to give domain definitions. |
| P4 L3-4: The 1st vertical layer seems quite deep. Where do you base such a choice?
Can you support this height for your region? | The lowest vertical layer thickness in of 50 m EMEP4UK was smaller than the standard value of 100 m in EMEP up to 2017. The model in this configuration has been shown to perform adequately in several previous studies (Vieno et al. 2010, Ots et al. 2016). In late 2017 the standard EMEP horizontal and vertical grid structures have changed to allow increased flexibility, however this change is not included in the current modelling. | A sentence has been added at the end of the first paragraph of section 2.2. |
| P4 L8: please provide reference(s) for the VBS scheme you use. | | Reference added to Bergström et al., 2012. |

| Referee comment | Author's Response | Changes in manuscript |
|---|---|---|
| P4 L5-12: which is the size distribution of aerosol species within the regional model? | Five classes of fine and coarse particles, with differing size and deposition properties, are used internally in EMEP4UK (Simpson et al. 2012 table 6), though inputs and outputs are given as PM2.5 and PM10. | Additional text added. |
| Sect. 2.3: Some important information for the city scale model is not given. i.e:
 a) which are the simulated pollutants (chemistry, sizes in case particles)?
 b) Which spatial criterion (coordinates) defines the simulation domain?
 c) Which is the simulation period of this model?
 d) Which is the temporal resolution of the simulations/outputs?
 e) The way/method used to provide ic/bc conditions to the city scale model is not described (pollutants, spatial distribution of used stations, temporal resolution etc). | | a) The final paragraph of section 2.3 has been extended to make this clearer.
 b) The Greater London simulation domain is defined by the LAEI emissions inventory extent.
 c) Description of meteorology data extended to clarify hourly data for full 2012.
 d) Clarified after boundary conditions description.
 e) Description of boundary conditions (upwind background concentrations) extended. |
| Sect. 2.4: the coupling method is quite unclear. Despite based on a previous study, it is better briefly described shortly here as well. | | The paragraph describing the concept of the coupled model to avoid double-counting emissions has been revised and expanded. |
| P6 L17-18: unclear. Please rephrase | | This explanation of the model version differences has been revised. |
| *it may be helpful to provide a table with the model applications used (incl. emission scenarios), for each purpose. | The authors have generally revised the text to make the different model configurations clearer and did not add another table to avoid excessive length. | |
| Sect. 3.1: the justification with respect to model errors is not quite understood (P9 L16-21) | | First paragraph of section 3.1 expanded to clarify. |
| Sect 3.2: not knowing the simulation period, it is a question how the annual average is calculated (e.g. from representative simulation events?) | All the models carry out hourly calculations and the annual averages are post-processed from the hourly concentrations. | First sentence of section 3.2 extended. |

| Referee comment | Author's Response | Changes in manuscript |
| --- | --- | --- |
| Sect. 4. I would rather prefer a balance between evaluation results and AQ results for the area of interest. At the moment, mostly the model performance is discussed rather than AQ issues. | The model evaluation has been put in the context of air pollutant dispersion and chemistry processes. Discussion has been added related to several additional aspects of air quality. | Additional discussion of vertical structure of concentrations added (refer to first comment of other referee). Discussion of increments of concentration between near-road and background sites added. Frequency scatter plots for NOx added with significant additional discussion in section 3.5. |
| Tables are quite many in number. Consider either merging those with similar information or moving the less important to the Appendix. | | Tables 6 and 7, containing model evaluation statistics for CO and PM, have been combined. |
| Figure 2: the quality of the figure seems poor. | The authors feel that this figure is adequate. Revisions to this figure could be made if the sub-editor considers them necessary for publication. | |
| Figure 4: the mass emission rates are not pronounced on the map. | | This figure has been recreated in different software with stronger colour contrasts. |
| Figures 5-7: the information on the model (system) used to produce these outputs is omitted. | | 'from the coupled model' added to the figure captions (also Figure A1). |
| Figure 9: the figure is poorly described in the legend. Bear in mind that each figure should stand alone. | | Additional explanation has been provided in the captions to Figures 9 and A4. |
| Figures 10- 11: the axis numbers of the middle plots are on top. | | The non-standard axis labels have been moved. |
| Figure 12: axis titles are duplicated. | | The axis titles have been altered. |

**Additional author response**

**Additional changes in manuscript**

A few minor typographical/grammar errors have been corrected.